# Fitting and comparison of calcium-calmodulin kinetic schemes to a common data set using non-linear mixed effects modelling

**Domas Linkevicius**[1], **Angus Chadwick**[1], **Guido C. Faas**[2], **Melanie I. Stefan**[3], **David C. Sterratt**[1] *

**1** Institute for Adaptive and Neural Computation, School of Informatics, University of Edinburgh, Edinburgh, United Kingdom, **2** Department of Neurology, School of Medicine, University of California Los Angeles, Los Angeles, California, United States of America, **3** Faculty of Medicine, Medical School Berlin, Berlin, Germany

* David.C.Sterratt@ed.ac.uk

**Data Availability Statement:** https://doi.org/10.6084/m9.figshare.c.7614467.v1.

## Abstract

Calmodulin is a calcium binding protein that is essential in calcium signalling in the brain. There are many computational models of calcium-calmodulin binding that capture various calmodulin features. However, existing models have generally been fit to different data sets, with some publications not reporting their training and validation performance. Moreover, there is no model comparison using a common benchmark data set as is common practice in other modeling domains. Finally, some calmodulin models have been fit as a part of a larger kinetic scheme, which may have resulted in parameters being underdetermined. We address these three limitations of previous models by fitting the published calcium-calmodulin schemes to a common calcium-calmodulin data set comprising equilibrium data from Shifman et al. and dynamical data from Faas et al. Due to technical limitations, the amount of uncaged calcium in Faas et al. data could not be predicted with certainty. To find good parameter fits, despite this uncertainty, we used non-linear mixed effects modelling as implemented in the `Pumas.jl` package. The Akaike information criterion values for our reaction rate constants were significantly lower than for the published parameters, indicating that the published parameters are suboptimal. Moreover, there were significant differences in calmodulin activation, both between the schemes and between our reaction rate and those previously published. A kinetic scheme with independent lobes and unique, rather than identical, binding sites fit the data best. Our results support two hypotheses: (1) partially bound calmodulin is important in cellular signalling; and (2) calcium binding sites within a calmodulin lobe are kinetically distinct rather than identical. We conclude that more attention should be given to validation and comparison of models of individual molecules.

## Introduction

Calmodulin is among the most important calcium binding proteins in the brain. It is essential in the translation of intracellular $Ca^{2+}$ signals to downstream processes, such as gene

**Funding:** Domas Linkevicius was funded by a PhD stipend by the United Kingdom Research and Innovation (grant EP/S02431X/1), UKRI Centre for Doctoral Training in Biomedical AI at the University of Edinburgh, School of Informatics. Guido C. Faas was funded by NIH grants NS027528 and NS030549 and The Wendy and Leonard Goldberg Endowment to UCLA. David C. Sterratt, Melanie I. Stefan and Angus Chadwick did not receive any specific funding for this work.

**Competing interests:** The authors have declared that no competing interests exist.

regulation, protein activation, metabolic regulation and synaptic plasticity [1, 2]. Calcium signals can provide information both via their amplitude (nanomolar to millimolar) and via their duration (microseconds to hours) [1, 2]. Calcium-calmodulin binding kinetics underlie the translation of $Ca^{2+}$ signals, therefore correct kinetic models of binding are an important aspect in studying calcium signalling in the brain.

Calmodulin's signalling properties arise from its structure—it comprises a 148 amino acid residue polypeptide with four EF hands divided into C and N lobes capable of binding two calcium ions per lobe [3]. It can adopt many conformational states, especially when bound to different molecules [4]. Moreover, calmodulin lobes have been reported to differ in their kinetics and affinity for $Ca^{2+}$–the N lobe binding faster with lower affinity and the C lobe binding slower with higher affinity [5–7].

There are at least 19 published computational models of synapses that include various models of calmodulin in their chemical reaction network [8]. The published kinetic schemes describing calcium-calmodulin binding vary significantly in the number of calmodulin features they capture. For example, some calmodulin models do not have independent lobes [9–11] while others do [5, 12, 13]. Some schemes are event-based—only concerned about the $Ca^{2+}$ binding events [10, 11], whereas others explicitly indicate which lobe and/or site is being bound to [5, 12, 13]. Moreover, some models assume that two $Ca^{2+}$ ions bind to a lobe at the same time [9, 10, 12], others leave this dependent on reaction rate constants [5, 11, 13]. Some models assume that $Ca^{2+}$ binding sites within a lobe are unique [11, 13] while others assume that they are non-unique [5, 12]. Finally, some models include details such as calmodulin conformational states [14].

Most current computational calmodulin models suffer from three limitations. First of all, different models have generally been tuned to different data sets, making their relative performance difficult to compare. Secondly, most models have not been cross-validated, making their generalization performance uncertain. Thirdly, some models have been tuned as a part of a larger scheme, e.g. including CaMKII, potentially making calcium-calmodulin binding parameters underdetermined. We discuss the sources of data to which calcium-calmodulin models we investigate have been fit in the Methods section, therefore we will next elaborate on the second and the third limitations.

The second limitation relates to cross-validation, a crucial step in the process of parameter inference used to establish model performance outside of the training data and to avoid overfitting (see page 241 in [15]). Ideally different models are cross-validated on a single data set across publications using consistent quantitative metrics. For example, the MNIST data set [16] is used to compare the error rate of image processing models. Given the lack of rich open access calmodulin data sets, none of the published calmodulin models were quantitatively cross-validated during development. At best, publications that contain calmodulin kinetic schemes include some indication (usually visual, rather than quantitative) of performance compared to the source of data they are being tuned to. However, this is not a rigorous way of ensuring that a model will perform well outside of the training data, leaving the generalization performance uncertain.

The third limitation is that some calmodulin schemes have been tuned to data from experiments that include other calmodulin-binding molecules, with large numbers of reaction rate constants that have to be fit, for example the calcium-calmodulin-CaMKII cascade [13]. Systems biology models naturally exhibit sloppiness [17], which tends to get more pronounced with an increasing number parameters being fit, resulting in loosely constrained parameter values. It is often possible to trade off between reaction rate constants: a calcium-calmodulin-CaMKII cascade being fit to CaMKII activity measurements may fit data better if $Ca^{2+}$ binds to calmodulin with higher affinity or calcium-calmodulin binds to CaMKII with higher

affinity, or some mix of the two. Moreover, a similar trade-off is possible between the binding sites and/or lobes even with only a calcium-calmodulin cascade. Because of these trade-offs, the true reaction rate constants might be completely different to the ones obtained via the fitting procedures. Some publications attempt to test for such parameter sloppiness via sensitivity analyses [18] or by calculating the eigenvalues of the Hessian [17], but it usually is too difficult to test the parameter combinations in a sufficiently dense and wide manner to ensure that the reaction rate constants are not under-determined.

Calmodulin models, in particular some of the simpler ones [9, 12], have been used to investigate calmodulin interactions with other molecules [18–20] and in complex chemical reaction networks [9, 21–23] to model higher order phenomena occurring in neurons, e.g. synaptic plasticity. However, given the aforementioned model limitations, it is important to scrutinize the previous modelling work, its basic assumptions, and to check whether the assumptions made in previous work hold when tested under more rigorous conditions, with powerful methods using richer data sets.

We address the three aforementioned limitations of existing calmodulin models by using a common data set where the only free kinetic parameters are calcium-calmodulin binding reaction rate constants. The common data set comprises subsets of data from Faas et al. [5] and Shifman et al. [11]. Faas et al. [5] contains time-series of fluorescence measurements after laser-induced $Ca^{2+}$ uncaging and therefore is informative about calmodulin dynamics. In contrast, Shifman et al. [11] contains measurements of calmodulin properties at equilibrium. To deal with incomplete experimental control of the amount of calcium uncaged by a laser flash in Faas et al. [5] we use the novel and highly efficient non-linear mixed effects (NLME) model fitting algorithms implemented in `Pumas.jl` [24]. NLME is a hierarchical modeling framework that can deal with phenomena where there are constant intra-individual parameters, but significant inter-individual variability due to individual level parameters [24, 25].

We use the common data set to fit reaction rate constants from scratch and compare our results to the reaction rate constants in the literature. By calculating the Akaike information criterion (AIC) [26] values for both our and the published reaction rate constants we show that the published reaction rate constants are suboptimal. Moreover, using the same criterion, we show that some kinetic schemes are suboptimal and fail to fit calmodulin dynamics and equilibrium behaviour at the same time. We then compare the $Ca^{2+}$ signal integration properties of different calmodulin schemes when either the published reaction rate constants or the ones determined by our approach are used. We show that there are significant differences in calmodulin calcium integration properties when using the suboptimal published reaction rate constants. Similarly, we show that the models using suboptimal calmodulin schemes display qualitatively different calcium integration behaviour compared to better performing schemes. Finally, we calculate the partial rank correlations between the reaction rate constants that we fit and show that for some calmodulin schemes our parameter fits are highly correlated which is indicative of parameter sloppiness or underdetermination.

Our results highlight that a sufficiently expressive calmodulin model structure is essential for capturing both calmodulin dynamics and equilibrium behaviour. Moreover, we conclude that, given the suboptimality of the previously published parameter sets, arguments and findings built on these models may warrant re-visiting.

## Methods

### Data

**Dynamical calmodulin data.**   We use the calcium uncaging data from Faas et al. [5], in which different concentration mixes of the fluorescent $Ca^{2+}$ indicator OGB-5N, the light

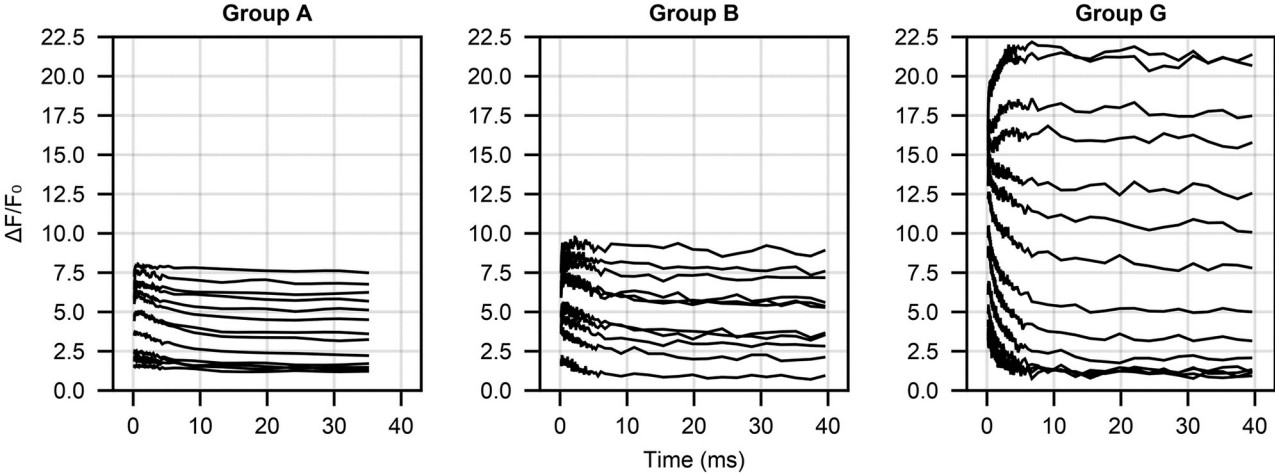

**Fig 1. Relative fluorescence ($\Delta F/F_0$) time-series data from [5] for three different initial condition groups (A, B and G).** Different lines within a plot are due to different laser uncaging strength (the higher the laser strength, the larger the $\Delta F/F_0$ value).

sensitive $Ca^{2+}$ chelator DM-nitrophen (DMn), calmodulin and titrated free $Ca^{2+}$ were used to make seven different groups of solutions A–G (see S2 Appendix for specific concentrations). For different batches of each group of solutions, a sequence of laser pulses of increasing strength was used to induce $Ca^{2+}$ uncaging from DMn while OGB-5N fluorescence was observed at 35˚C. The stronger the pulse, the more calcium is released. Due to technical limitations, it was not possible to predict the amount of released calcium for each laser pulse precisely. We elaborate on how we model the fraction of uncaged $Ca^{2+}$ below. Fig 1 shows the fluorescence time courses for three of the seven groups—A, B and G—and different uncaging laser strengths. We use a subset of the data and split it into training, validation and test data sets (see Supplemental Text S1 Appendix for more information).

**Calmodulin equilibrium data.** Steady-state calcium-calmodulin binding came from an experiment in Shifman et al. [11], which measured the number of $Ca^{2+}$ ions bound per calmodulin molecule at different free $Ca^{2+}$ concentrations (their Fig 1B). Their experimental chamber contained a fluorescent indicator Fluo4FF (5μM), calmodulin (5μM) and a varying amount of free $Ca^{2+}$. The amount of free $Ca^{2+}$ was titrated until a required concentration (between approximately $10^{-7}$M and $5.5 \times 10^{-5}$M) was reached. We used a digital tool (https://automeris.io/WebPlotDigitizer/) to extract this data from their plots, giving the 107 points shown in Fig 2. This data was obtained at 25˚C but calmodulin does not show significant temperature dependent changes in equilibrium behaviour [27], so we do not adjust for temperature dependent changes in calmodulin kinetics.

## Kinetic schemes and published reaction rate constants

We investigated six different calcium-calmodulin binding schemes from the literature that span the complexity of the most commonly used calmodulin models (Fig 3). The reference we give for a scheme may be its original source, or a source that is frequently cited for the scheme. There are more complex published calmodulin schemes that we did not use [14], because they would be prohibitively computationally expensive to fit.

The simplest scheme (Scheme 1, Fig 3), from Kim et al. [9], is made up of three calmodulin states—CaM0, CaM2Ca, CaM4Ca, respectively calmodulin bound to no, two and four $Ca^{2+}$ ions. In Scheme 1, $Ca^{2+}$ binding is assumed to be highly co-operative and binding of two $Ca^{2+}$

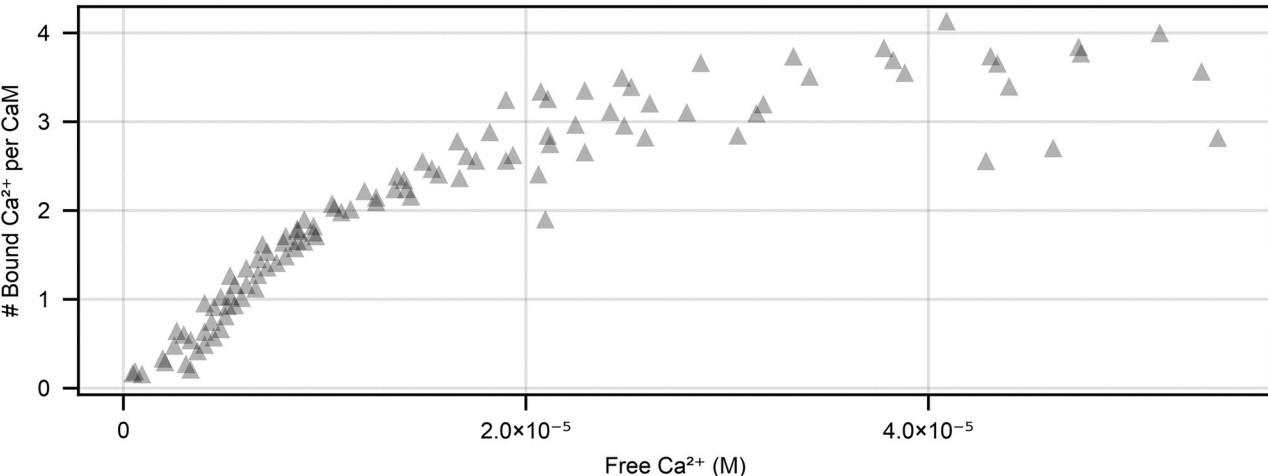

**Fig 2. Equilibrium measurements of the number of Ca²⁺ ions per calmodulin molecule from [11].** Experiments were done using 5μM Fluo4FF and 5μM calmodulin.

ions is treated as a single reaction. In principle Scheme 1 does not assume which lobe binds first; the first two Ca²⁺ ions could bind to the C lobe or the N lobe. However, the parameterisation of Scheme 1 by Kim et al. [9] implies that they treat the first Ca²⁺ binding event as being to the C lobe. The published reaction rate constants in Kim et al. [9] are based on stopped-flow fluorescence measurements [7]. Scheme 1 is parameterised by four reaction rate constants $\{k_i\}_{i=1}^4$, which can be used to derive two dissociation constants: $K_{D_1} = \frac{k_2}{k_1}$ and $K_{D_2} = \frac{k_4}{k_3}$.

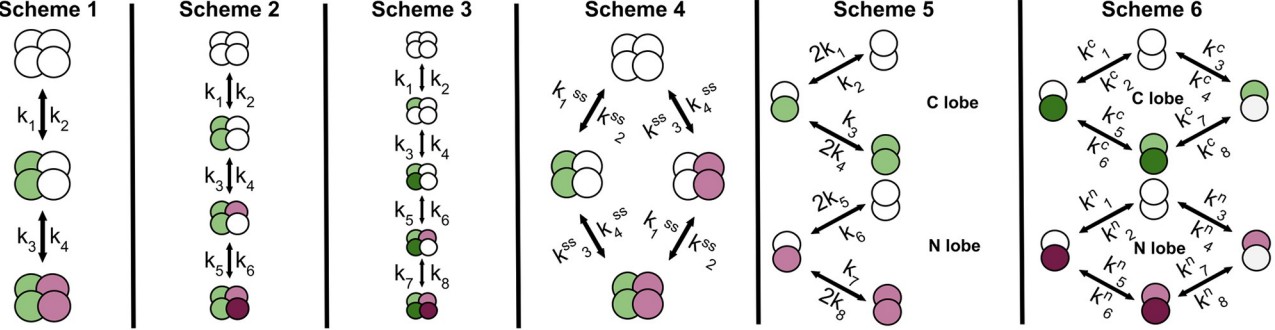

**Fig 3. Six calmodulin kinetic schemes to which we fit parameters, and compare to performance with published parameter values. Scheme 1**—due to strong co-operativity, each calmodulin lobe binds two Ca²⁺ ions at a time, with the lobes modelled sequentially (first C lobe then N lobe). Scheme 1 is parameterised by reaction rate constants $\{k_i\}_{i=1}^4$. **Scheme 2**—due to co-operativity the first reaction has two Ca²⁺ ions binding as a single event and then the next two Ca²⁺ ions binding sequentially. It is parameterised by reaction rate constants $\{k_i\}_{i=1}^6$. **Scheme 3**—fully expanded sequential calmodulin scheme where each binding event is represented individually. Depending on the reaction rate constants, the binding events could be mixed between the lobes, e.g. first binding event could be in the C lobe, the second in the N lobe, or partial combinations of different lobes. The visualised scenario is where the first two events are in the C lobe. This scheme is parameterised by reaction rate constants $\{k_i\}_{i=1}^8$. **Scheme 4**—calmodulin binds two Ca²⁺ ions at a time and, contrary to **Schemes 1–3**, the lobes are independent. It is parameterised by eight reaction rate constants $\{k_i\}_{i=1}^8$ which, along with free Ca²⁺, are used to calculate the effective reaction rate constant $\{k_i^{ss}\}_{i=1}^4$ (see S3 Appendix for more details). **Scheme 5**—this scheme has independent N and C lobes, with a single Ca²⁺ ion binding at a time. Binding sites within a single lobe are identical. It is parameterised by reaction rate constants $\{k_i\}_{i=1}^8$. **Scheme 6**—this scheme has independent N and C lobes, with a single Ca²⁺ ion binding at a time. In contrast to Scheme 5, the binding sites within a single lobe are distinct (indicated by different shades of green/purple). The scheme is parameterised by 16 reaction rate constants $\{k_i^{c/n}\}_{i=1}^8$. In all schemes green circles indicate Ca²⁺-bound C lobe sites, purple circles indicate Ca²⁺-bound N lobe sites and arrows indicate bidirectional reactions (Ca²⁺ ions not shown).

The next scheme (Scheme 2, Fig 3), from Bhalla and Iyengar [10], is made up of four calmodulin states—CaM0, CaM2Ca, CaM3Ca, CaM4Ca, respectively calmodulin bound to no, two, three and four $Ca^{2+}$ ions. In Scheme 2, binding of the first two $Ca^{2+}$ ions is assumed to be highly co-operative and treated as a single reaction, whereas the next two $Ca^{2+}$ ions bind individually. The parameters in Bhalla and Iyengar [10] (as given in https://doqcs.ncbs.res.in/, also see [28]) do not match neatly to either lobe and the description of how the rate constants were derived was unavailable at the time of writing. Scheme 2 is parameterised by six reaction rate constants $\{k_i\}_{i=1}^6$, which can be used to derive three dissociation constants: $K_{D_1} = \frac{k_2}{k_1}$, $K_{D_2} = \frac{k_4}{k_3}$ and $K_{D_3} = \frac{k_6}{k_5}$.

The final linear scheme that ignores calmodulin lobe-based structure (Scheme 3, Fig 3) is from Shifman et al. [11]. It comprises five calmodulin states—CaM0, CaM1Ca, CaM2Ca, CaM3Ca, CaM4Ca—respectively calmodulin bound to no, one, two, three and four $Ca^{2+}$ ions. The dissociation constants based on experiments in Shifman et al. [11] are 7.9μM, 1.7μM, 35μM, 8.9μM respectively for $Ca^{2+}$ binding events one to four. Reactions in this scheme do not neatly map onto individual $Ca^{2+}$ binding sites within calmodulin lobes; instead they are abstract binding events where, depending on the parameters, they may be probabilistic combinations between different binding sites. Scheme 3 is parameterised by eight reaction rate constants $\{k_i\}_{i=1}^8$, which can be used to derive four dissociation constants $K_{D_1} = \frac{k_2}{k_1}$, $K_{D_2} = \frac{k_4}{k_3}$, $K_{D_3} = \frac{k_6}{k_5}$ and $K_{D_4} = \frac{k_8}{k_7}$. This scheme is the most complex linear CaM scheme possible (without adding conformational calmodulin changes), modelling each $Ca^{2+}$ binding site individually.

Our Scheme 4 (Fig 3) is from Pepke et al. [12] and comprises four states—CaM0, CaM2C, CaM2N, CaM4Ca—respectively calmodulin bound to no $Ca^{2+}$ ions, two at the C lobe, two at the N lobe and four across both lobes. It is the simplest scheme that captures the lobe-based structure of calmodulin. It has eight reaction rate constants $\{k_i\}_{i=1}^8$ and is based on Scheme 5 (described below), but was simplified used a quasi-steady state approximation for calmodulin species that have a single bound $Ca^{2+}$ ion. This approximation results in elimination of partially bound species from simulations by setting their derivatives to 0 and expressing the partially bound species in terms of the unbound and the fully bound species and permitting the appropriate substitutions in the equations for the unbound and the fully bound species (see S4 Appendix).

We draw our Scheme 5 (Fig 3) from model 1 in Pepke et al. [12], which is identical to the scheme used in [5]. It is made up of nine states—CaM0, CaM1C, CaM1N, CaM2C, CaM2N, CaM1C1N, CaM2C1N, CaM1C2N, CaM4—with the number of $Ca^{2+}$ ions bound to calmodulin indicated by numbers preceding C and N. Even though in total there are nine states, since in this study calmodulin does not bind to any downstream species, we do not need to track individual calmodulin molecules. Therefore, we simulate the lobes as independent species which decreases the number of states we need to track from nine to six—CaM0N, CaM1N, CaM2N, CaM0C, CaM1C, CaM2C—without changing the scheme itself. Scheme 5 is parameterised by eight reaction rate constants $\{k_i\}_{i=1}^8$, which can be used to derive four dissociation constants $K_{D_1} = \frac{k_2}{k_1}$, $K_{D_2} = \frac{k_4}{k_3}$, $K_{D_3} = \frac{k_6}{k_5}$ and $K_{D_4} = \frac{k_8}{k_7}$. Pepke et al. [12] used two data sources on calmodulin equilibrium behavior: (1) data from wild-type and tryptic calmodulin fragments (one lobe expressed, other eliminated) [6]; (2) data from competition assays (calmodulin, either wild type or mutants with one active and one inactive lobe, and fluorescent indicator Fluo4FF) [11]. Pepke et al. [12] (in their supplemental information) give reaction rate constants as ranges—we take specific numerical values from this model's entry (model identifier: MODEL1001150000) in the BioModels Database [29, 30]. Faas et al. [5] tuned the model to their own UV-flash photolysis data.

Finally, our Scheme 6 (Fig 3) is from Byrne et al. [13]. It is made up of sixteen states, but similar to Scheme 5, we simulate the lobes as independent species which reduces the number of states to eight—CaM0N, CaMN$_1$, CaMN$_2$, CaM2N, CaM0C, CaMC$_1$, CaMC$_2$, CaM2C, where CaM0X denotes an unbound calmodulin lobe, CaMX$_1$ denotes Ca$^{2+}$ bound to the first site of a lobe, CaMX$_2$ denotes Ca$^{2+}$ bound to the second site of a lobe and CaM2X denotes a fully bound lobe. This scheme is parameterised by 16 reaction rate constants $\{k_i^{c/n}\}_{i=1}^8$, which can be used to derive eight dissociation constants $K_{D_1}^{c/n} = \frac{k_2^{c/n}}{k_1^{c/n}}$, $K_{D_2}^{c/n} = \frac{k_4^{c/n}}{k_3^{c/n}}$, $K_{D_3}^{c/n} = \frac{k_6^{c/n}}{k_5^{c/n}}$ and $K_{D_4}^{c/n} = \frac{k_8^{c/n}}{k_7^{c/n}}$. Reaction rate constants in Byrne et al. [13] are based on stopped-flow fluorescence and competitive binding assay data [31].

For each of the six schemes we use the reaction rate constants from the associated publication (we use both Pepke et al. [12] and Faas et al. [5] for Scheme 5). All of the reaction rate constants we used are given in S4 Appendix.

## Ca$^{2+}$ uncaging model

Faas et al. [5] used a linear model for laser induced Ca$^{2+}$ uncaging

$$U(\text{PCD}, x) = 0.0011 \times \text{PCD} - 0.39 + x \tag{1}$$

where $U$ is the uncaged DMn fraction, PCD is the specified Pockels cell delay, where a larger value results in a higher energy laser pulse and more Ca$^{2+}$ uncaging, and $x$ is used to account for the uncertainty of the actual PCD value, i.e. the difference between the specified and physically realised values.

Even though Faas et al. [5] used the linear model successfully in their study, its performance is not quantified and it has some limitations. Most importantly, $U$ is not bounded to $[0, 1]$ and can take negative values or values above 1, which would result in physically unrealistic initial conditions. Moreover, it is not clear that a simple linear relationship is optimal to accurately model the relationship between the PCD value and the uncaging fraction. Finally, the variable $x$ is additive, and it is not clear that this formulation is optimal—it could be multiplicative or some more complex functional relationship.

Due to lack of the necessary data, we could not develop our own model of how the fraction of calcium uncaged depends on the PCD. Instead, of the linear model (Eq 1) we use an even simpler model that performed better in practice than either the linear model from Faas et al. [5] or a neural network. Our uncaging model does not take the specified PCD value and is a simple sigmoid function, bounding its output to $[0, 1]$.

$$U(x) = \frac{1}{1 + \exp^{-x}} \tag{2}$$

With this approach, we do not claim that uncaging is completely independent of PCD; rather we use a single equation to capture both uncertainty in estimating the PCD, as well as other sources of variance.

### Fitting our reaction rate constants

We combine and adapt the definitions and notation of NLME provided in [24, 32] and we present it using Scheme 1 as an example. NLME modeling framework comprises a two level hierarchical structure (shown visually in Fig 4) with fixed effects $\Theta$ at the upper level, which can be broadly grouped into

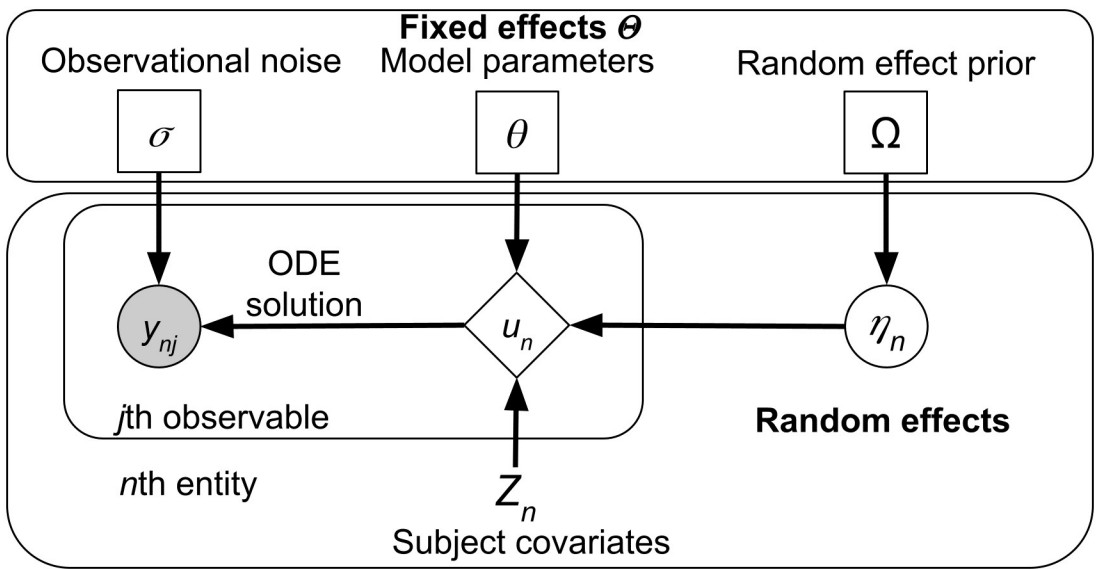

**Fig 4. Visual representation of an NLME model, rectangle nodes in the top box denote parameters (fixed effects), circles denote random quantities which are either latent (unfilled) or observed (filled), diamonds are deterministic given the inputs, and nodes without a border are constant.** Each symbol in the node can be either a number or a vector depending on the context.

- model parameters $\theta$, e.g. for scheme 1—reaction rate constants $\{k_i\}_{i=1}^{4}$

- random effect prior distribution parameters $\Omega$, e.g. $\mu$ and $\omega$ used to parameterize the prior of $\eta_n$ for the $Ca^{2+}$ uncaging fraction (see paragraph below)

- observation model noise parameters $\sigma$

and do not vary between recordings.

The lower level is random effects $\eta_n$ which account for the inter-individual variability of the observations $y_{nj}$, generally, by individualizing model parameters $\theta$. We assume that both $Ca^{2+}$ and calmodulin molecules are identical between experiments, hence reaction rate constants do not have the random effect-enabled individualization for each experimental run. The sole usage of random effects in this paper is in fitting the fraction of uncaged $Ca^{2+}$ by passing $\eta_n \sim \mathcal{N}(\mu, \omega)$ to Eq 2.

Furthermore, there is a set of covariates $Z_n$ associated with each recording, i.e. the total concentration of calmodulin, $Ca^{2+}$ and OGB5N, which are known. These three sets of values are collated via the parameter model $g$ into the dynamical parameter vector $p_n$ of the $n$th recording

$$p_n = g(\Theta, Z_n, \eta_n) \tag{3}$$

The dynamical parameters $p_n$ are then fed into the structural model (e.g. an ordinary differential equation (ODE) system)

$$u'_n = f(u_n, p_n, t) \tag{4}$$

where $u$ are the dynamical variables being solved for (DMn, OGB5N, $Ca^{2+}$ and their combinations and the various calmodulin species determined by the scheme being used). For Scheme 1 the system of equations (for brevity omitting DMn, OGB5N, $Ca^{2+}$ and means of its input

amplitude, frequency and duration into the model) would be

$$
\begin{aligned}
\mathrm{d}[\mathrm{CaM0}]/\mathrm{dt} &= -k_1[\mathrm{Ca}]^2[\mathrm{CaM0}] + k_2[\mathrm{CaM2Ca}] \\
\mathrm{d}[\mathrm{CaM2Ca}]/\mathrm{dt} &= k_1[\mathrm{Ca}]^2[\mathrm{CaM0}] - k_2[\mathrm{CaM2Ca}] \\
&\quad - k_3[\mathrm{Ca}]^2[\mathrm{CaM2Ca}] + k_4[\mathrm{CaM4Ca}] \\
\mathrm{d}[\mathrm{CaM4Ca}]/\mathrm{dt} &= k_3[\mathrm{Ca}]^2[\mathrm{CaM2Ca}] - k_4[\mathrm{CaM4Ca}]
\end{aligned}
\tag{5}
$$

Note that $\{k_i\}_{i=1}^4$ enter into Eq 4 via $p_n$, which can also be used to initialize the ODE system.

The final step is to link the numerical solution of the ODE system to the experimentally observed quantities. The $j$th observable quantity $y_{nj}$ for the $n$th entity is calculated using the simulated variables $u_n(t)$ and the times $t_m$ at which the observations were made via the observational model $h$

$$
y_{nj}(t = t_m) = h_j(u_n(t = t_m), p_n, Z_n, \eta_n)
\tag{6}
$$

In this study there are two observable quantities: the relative fluorescence $\Delta F/F_0$ over time being fit to recordings from Faas et al. [5] and $\mathrm{Ca}^{2+}$ per calmodulin at equilibrium being fit to Shifman et al. [11]. $\Delta F/F_0$ is derived from OGB5N as follows

$$
\Delta F/F_0(t = t_m) = \frac{[\mathrm{OGB5N}](t = t_m) + (F_{\max}/F_{\min})[\mathrm{CaOGB5N}](t = t_m)}{[\mathrm{OGB5N}](t = 0) + (F_{\max}/F_{\min})[\mathrm{CaOGB5N}](t = 0)}
\tag{7}
$$

where $F_{\max}/F_{\min} = 39.364$ [5] and $F_{\max}$, $F_{\min}$ are maximal and minimal recorded fluorescence values. $\mathrm{Ca}^{2+}$ per calmodulin is simply the sum of calmodulin species multiplied by the number of bound $\mathrm{Ca}^{2+}$ ions for each species divided by total calmodulin. After obtaining the observable quantities a Gaussian observation model is used to account for observational noise.

There are many ways to fit NLME models, both frequentist and Bayesian [33]. In this study we use the maximum aposterior (MAP) conditional log-likelihood objective which can be stated as

$$
\Theta^*, \eta^* = \arg\max_{\Theta, \eta} \left( p(\Theta) \cdot \prod_{i=1}^{N} p(y_n \mid \Theta, \eta_n) \cdot p(\eta_n \mid \Theta) \right)
\tag{8}
$$

where $\Theta^*$ is the mode of the fixed effects, $\eta^*$ is the mode of the random effects for each subject and $p(\Theta)$ is the fixed effect prior distribution. Conditional likelihood is much more numerically efficient due to $\Theta$ and $\eta_n$ being optimized jointly whereas, for example, marginal likelihood generally requires a two level optimization scheme and Markov Chain Monte Carlo requires many more likelihood evaluations due to sampling. However, conditional likelihood requires appropriate handling (either fixing or priors, see next section) of $\Omega$ to avoid overly broad random effect distributions which barely penalize extreme $\eta_n$ values and effectively result in different individual models due to the learning being offloaded mostly to the random effects.

We use the Pumas.jl [24] Julia package to fit to solve Eq 8. Pumas.jl contains efficient and powerful algorithms for NLME modelling, which was essential when fitting the $\eta$s used to model the uncaging fraction. Specifically, we used the BFGS optimization algorithm from Optim.jl with the gradient calculations handled by Pumas.jl.

All fitting was done on the JuliaHub (https://juliahub.com/) cloud computing platform using nodes with 8 vCPUs and 64GB of memory. Individual fits took between one to ten minutes, depending on which scheme was used and whether some of the parameters were fixed.

For each scheme we conducted 20 fitting runs with different initial conditions. All the code that was used to define the models, run the simulations and perform the analysis is accessible at https://github.com/dom-linkevicius/FaasCalmodulin.jl.git, the data is accessible at [34].

## Prior distributions

We incorporated the existing knowledge about calmodulin reaction rate constants for different kinetic schemes via per-scheme prior distributions that depend on the amount data available for each scheme. All of our priors are in $\log_{10}$ space as optimizing rate constants in log-space was more performant.

For Schemes 1 and 2, since they are simplified and contain fewer reaction rate parameters than an actual calmodulin molecule would, mapping from experimental data to reaction rate constants is difficult. Therefore, we opted to use wide uniform priors that reflects the small amount of available prior information: $\mathcal{U}(2, 9)$ for the forward reaction rate constants (corresponds to $10^2$ $M^{-2}ms^{-1}$ to $10^9$ $M^{-2}ms^{-1}$) and $\mathcal{U}(-9, -4)$ for the dissociation constants (corresponds to a range of 1 $nM^2$ to 10 $\mu M^2$).

For Scheme 3, which models $Ca^{2+}$ binding events individually, there is a significant amount of prior information. Specifically, we use set priors on the dissociation constants based on Shifman et al. [11]. We used priors of the form $\mathcal{N}(r, 1)$, where $r$ is a dissociation constant from Shifman et al. [11] Table 2 in $\log_{10}$. Unfortunately, setting a prior that could similarly constrain the forward reaction rate constants was not possible, therefore we again opted for wide uniform priors that we used for Schemes 1 and 2: $\mathcal{U}(2, 9)$.

For Schemes 4 and 5, since they share the same set of reaction rate constants, we used the same set of prior distributions. For each forward reaction rate and dissociation constant we used $\mathcal{N}(r, 1)$, where $r$ are reaction rate constants from Faas et al. [5] in $\log_{10}$. We chose Faas et al. [5], rather than Pepke et al. [12], because their rate constants are based on dynamical data and upon initial simulation runs were performing better. Similarly for Scheme 6, we used the same approach, but centered the Gaussian priors on parameters from Byrne et al. [13].

Finally, we restrict the values of $\mu$ and $\omega$ which parameterize the prior distribution of random effects $\eta_n \sim \mathcal{N}(\mu, \omega)$ to avoid over-fitting due to the usage of the MAP conditional likelihood as the optimization objective. Specifically, we use $p(\omega) = \mathcal{N}(0, 1)$ and limit its domain to $[1, \infty]$, as well as limiting the domain of $\mu$ to be in $[-5, 5]$. We found that these were the minimal set of restrictions that prevented over-fitting of $\Omega$.

## Numerical ODE solving

We use the Julia programming language for numerical ODE solving both during and outside of parameter fitting. Specifically, we use the `DifferentialEquations.jl` package [35]. We use the `Rodas5P` numerical solver which can handle significant stiffness in the ODE system and which performed the best of the methods tried. We used it with the default settings, except for reducing the absolute error tolerance to `abs_tol = 1e-16` since some simulations that contained low concentrations of species suffered from significant errors in the numerical solution.

## Model comparison

There are many ways to compare model performance, but for the purposes of this study we use two metrics: root-mean-square error (RMSE) and the Akaike information criterion (AIC) [26]. The RMSE for a single experimental observation vector $y_{nj} \in \mathbb{R}^m$ and model prediction

$y_{nj}^* \in \mathbb{R}^m$ is defined as

$$\text{RMSE}(y_{nj}, y_{nj}^*) = \sqrt{\sum_{i=1}^{m} (y_{nj,i} - y_{nj,i}^*)^2} \tag{9}$$

We calculated the RMSE values for each recording in each of the test data sets for each of the 20 optimization runs for a given scheme, pooling them. We also calculated the RMSE values for the same scheme but with published reaction rate constants, again pooling them. This gave us two samples of RMSE values, $r_1$ that was obtained using the reaction rate constants we fitted and $r_2$ that was obtained using the published reaction rate constants. We then compared $r_1$ and $r_2$ using a two sample T-test (assuming unequal variance) and calculated Cohen's $d$ to establish the effect size of using our rates and the ones published in the literature. Cohen's $d$ is defined as

$$d = \left| \frac{\bar{r}_1 - \bar{r}_2}{s} \right| \tag{10}$$

where $\bar{r}_1$ and $\bar{r}_2$ are the means of each sample and $s$ is the pooled variance. We used the RMSE to focus directly on a models predictive performance.

In contrast, we used the AIC for selecting the model that performs the best when its complexity is taken into account. The AIC for model $\mathcal{M}$ with parameters $\theta$ and given data $d$ is defined as

$$\text{AIC}(d, \mathcal{M}, \theta) = 2k - 2\mathcal{L}(d|\mathcal{M}, \theta) \tag{11}$$

where $k$ is the length of the parameter vector $\theta$ (in this paper—reaction rate constants for a particular scheme, noise parameter $\sigma$ and random effect prior parameters $\mu$ and $\omega$). Even though in model optimization we use the conditional likelihood, in AIC calculations we used the marginal likelihood $\mathcal{L}$ obtained via the Laplace approximation [36]. The AIC is a measure that evaluates model performance, but also penalizes model complexity via the $2k$ term. There are many other model comparison metrics [37], but the AIC is sufficient for the present study due to the inclusion of predictive model performance and penalizing model complexity along with it being computationally simple to calculate. For each of the 20 different optimization runs we calculated the AIC value for the test data set of a run using the given scheme with our reaction rate constants, as well as if the published reaction rate constants were used. This gave us one sample of AIC values per combination of scheme + reaction rate constants.

## Results

### General model fitting results

We fit each of the six kinetic schemes shown in Fig 3 to the fluorescence traces from Faas et al. [5] and the steady state calcium-calmodulin binding data from Shifman et al. [11]. We used the root mean square error (RMSE) to evaluate the goodness of fit between the models and the data. The fitting procedure was repeated 20 times with different random seeds, which set the random sampling of the training, validation and test data (see S1 Appendix) along with the initial parameters for optimization. Therefore, due to variability in training data and initial parameters, RMSE values (especially for our parameter fits) for each seed can be significantly different.

We now compare the performance of each kinetic scheme with the reaction rate constants we fit and the published ones. Table 1 shows the training (split between dynamical data from Faas et al. [5] and equilibrium data from Shifman et al. [11]), validation and test data set

**Table 1. Summary of training, validation and test performance (RMSE ± SD) for different kinetic schemes with either parameters fit from scratch, fixed to values from publications or our modifications.**

| Scheme + rate constants | Training (Dynamical data) | Training (Equilibrium data) | Validation (Dynamical data) | Testing (Dynamical data) | Cohen's d[‡] |
|---|---|---|---|---|---|
| Scheme 1 + our fits | 0.74 ± 0.12 | 1.47 ± 0.17 | 0.80 ± 0.16 | 0.77 ± 0.17 | - |
| Scheme 1 + Kim et al. | 1.51 ± 0.20 | 2.31 | 1.16 ± 0.06 | 1.11 ± 0.08 * | 2.07 |
| Scheme 2 + our fits | 0.72 ± 0.09 | 2.00 ± 0.28 | 0.85 ± 0.15 | 0.81 ± 0.17 | - |
| Scheme 2 + Bhalla and Iyengar | 1.03 ± 0.04 | 2.23 | 1.16 ± 0.06 | 1.11 ± 0.08 * | 1.77 |
| Scheme 3 + our fits | 0.43 ± 0.03 | 0.83 ± 0.46 | 0.48 ± 0.05 | 0.46 ± 0.05 | - |
| Scheme 3 + Shifman et al.[†] | 0.61 ± 0.05 | 0.46 | 0.70 ± 0.11 | 0.66 ± 0.10 * | 3.76 |
| Scheme 4 + our fits | 0.42 ± 0.06 | 0.88 ± 0.10 | 0.46 ± 0.12 | 0.44 ± 0.11 | - |
| Scheme 4 + Pepke et al. | 0.82 ± 0.05 | 0.78 | 0.84 ± 0.06 | 0.82 ± 0.09 ** | 3.47 |
| Scheme 5 + our fits | 0.37 ± 0.02 | 0.44 ± 0.18 | 0.40 ± 0.03 | 0.38 ± 0.04 | - |
| Scheme 5 + Faas et al. | 0.45 ± 0.02 | 0.75 | 0.56 ± 0.03 | 0.53 ± 0.03 ** | 4.17 |
| Scheme 5 + Pepke et al. | 0.86 ± 0.05 | 0.75 | 0.88 ± 0.07 | 0.86 ± 0.09 ** | 13.5 |
| Scheme 6 + our fits | 0.35 ± 0.02 | 0.35 ± 0.03 | 0.38 ± 0.03 | 0.36 ± 0.03 | - |
| Scheme 6 + Byrne et al. | 0.47 ± 0.01 | 0.83 | 0.55 ± 0.03 | 0.53 ± 0.02 ** | 5.83 |

Highlighted rows were the best performing for that scheme. Note that for models with fixed reaction rate constants trained on Shifman et al. [11] data, no SD is given since all of the data is used for training and there is no variance in this metric. T-tests are done with reference to our reaction rate constants.

[†] Shifman et al. [11] only contained dissociation constant values, therefore we had to fit on-rate constants while fixing to their $K_D$ values.

[‡] Effect sizes evaluated via Cohen's d over 1.2 are considered very large and over 2.0 are considered huge [38].

* $p < 10^{-6}$

** $p < 10^{-10}$

performance summary statistics for the six investigated kinetic schemes, reporting the average RMSE values for the 20 different seeds used. The distribution of the test RMSE values is shown in Fig 5. On average Schemes 5–6 performed the best, whereas other schemes were not able to capture either the equilibrium data (Schemes 3–4) or both the dynamical and the equilibrium data (Schemes 1–2).

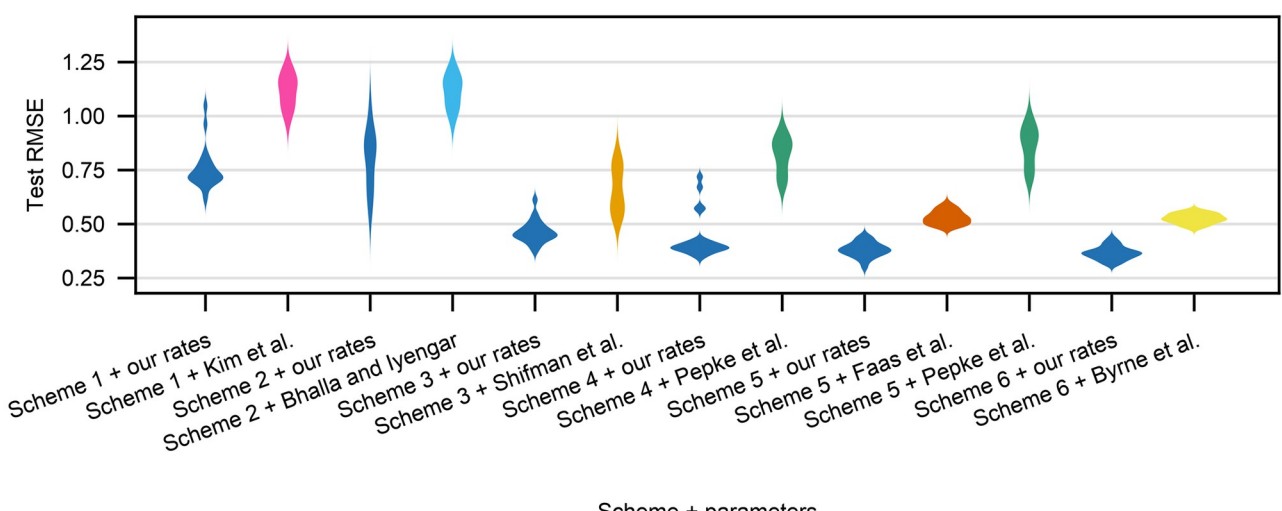

**Fig 5. Violin plots of RMSE values for the test data set for each seed for all schemes for our own and the published parameter sets.**

**Dynamical behaviour.** To illustrate the differences between model fits, Fig 6 shows the measured fluorescence traces that were used as the validation data set for seed 1 and corresponding model predictions for each kinetic scheme with our fitted rate constants and the published rate constants. Each row shows the same 24 experimental traces (black), split between the seven groups of solutions which were used experimentally in Faas et al. [5]. Within each group, each trace corresponds to a different laser uncaging strength used—the stronger the laser, the more calcium gets released, the larger the $\Delta F/F_0$ values that are measured.

The biggest differences in measured and predicted dynamics are for Scheme 1, for which the RMSE differences are also the largest. In Fig 6 the main difference between our parameters and rate constants from Kim et al. [9] is that our rate constants give rise to traces that follow the experimental dynamics to some extent, whereas the published rate constants simply equilibrate to a value and barely display any dynamics (e.g. groups D–G). However, even though our rate constants result in dynamical behaviour, they do not show good equilibrium performance and in fact do not reach equilibrium when the data has long reached it.

The comparison of dynamics for Scheme 2 is similar to that of Scheme 1. Comparing our reaction rate constants with those in Bhalla and Iyengar [10], we see that the published reaction rate constants make the system equilibrate and not follow the data closely, whereas our reaction rate constants achieve a more accurate fit. However, with either our rate constants or the published rate constants, the traces and the average RMSE values indicate that Scheme 2 fits the data quite poorly.

Models with the level of complexity of Scheme 3 and higher are able to capture the dynamical data much better than the simpler Schemes 1 and 2. As shown in Fig 6, both Scheme 3 models perform adequately. However, our reaction rate constants trained from scratch still perform better, especially in capturing the initial rise and fall in $\Delta F/F_0$ (for example see columns D–F). Note that for this scheme the comparison is not entirely equivalent to other cases as we had to fit the forward reaction rate constants while we kept the dissociation rate constants fixed to those in Shifman et al. [11].

For Scheme 4, our reaction rate constants significantly outperform those of Pepke et al. [12] as shown in Fig 6. This is reflected in a smaller mean RMSE value of our rate constants and is evident in most experimental groups, where our rate constants result in reasonably accurate predictions whereas the Pepke et al. rate constants significantly under-predict $\Delta F/F_0$.

Looking at the dynamics for Scheme 5, based on the RMSE values in Table 1, our rate constants perform significantly better than the rate constants from either Faas et al. [5] or Pepke et al. [12], but the gap is much smaller for the former than the latter. The differences in dynamics between our fits and rate constants in Faas et al. are subtle, but generally our rate constants perform better for small amounts of uncaged $Ca^{2+}$. In contrast, comparing dynamics with our rate constants to dynamics with rate constants in Pepke et al. [12], their reaction rate constants result in significant mismatches to the data, to the point that the optimization procedure has to inject amounts of $Ca^{2+}$ that lead to incorrect equilibrium levels (see Fig 7, which shows that their dissociation constants can fit equilibrium data well).

Finally, for Scheme 6, the main differences between the dynamics resulting from our reaction rate constants and those in Byrne et al. [13] are generally seen for small amounts of uncaged calcium (columns D–G bottom traces). Our reaction rate constants (for this seed) managed to capture calmodulin behaviour with low amounts of $Ca^{2+}$ more accurately. Even though for some traces the published rate constants can outperform ours (e.g. top traces in either groups A or C), our reaction rate constants on average show a smaller RMSE value.

**Equilibrium behaviour.** Fig 7 shows the equilibrium behaviours of our reaction rate constants (for all 20 training seeds) and published ones. When using Scheme 1 (Fig 7, top row),

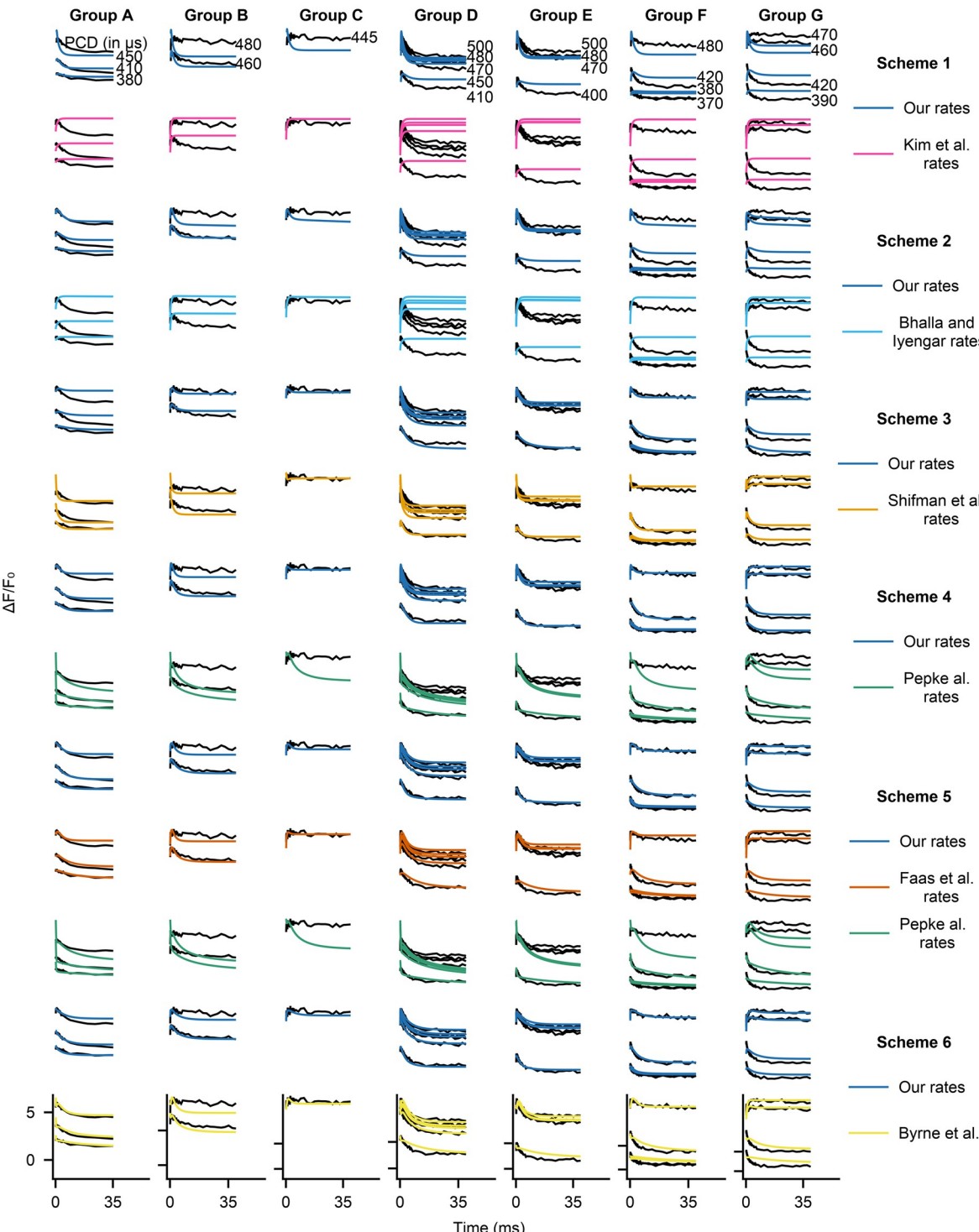

**Fig 6. Sample dynamics for validation data for seed 1 for all trained models.** Each column is a single data group A–G from Faas et al. [5], whereas each row is a different combination of scheme + reaction rate constants (specific combination given in the rightmost column). In all subplots y axis is $\Delta F/F_0$ and x axis is time. Black lines are empirical data and red lines are model outputs. Scale bars are given at the bottom row, each tick on the y-axis corresponds to the same value in each column.

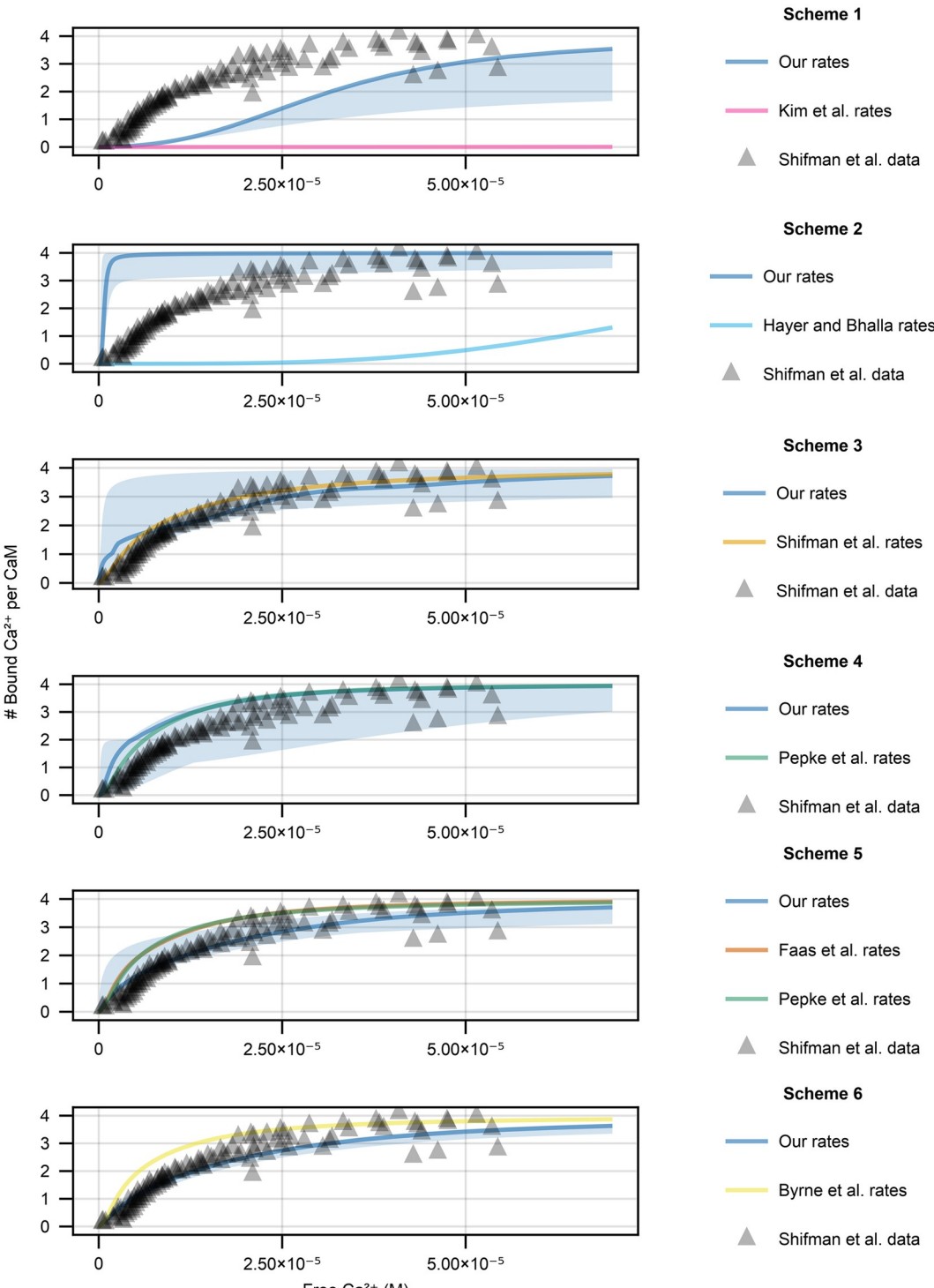

**Fig 7. Calmodulin equilibrium behaviour when the free amount of Ca$^{2+}$ is varied for all kinetic schemes for the 20 seeds that we tested (solid line is the median and shaded area is the 95% confidence interval).** Model behaviour with our reaction rate constants are plotted in blue, whereas published ones are in the colors indicated. Note that at times the published reaction rate constants (Scheme 1 and Scheme 2) result in behaviours that are much more right-shifted, therefore show up as zero in the relevant range. We also include data points for wild type calmodulin for an equivalent experimental setup in Shifman et al. [11].

the fits to data from Shifman et al. [11] are generally poor. A single run was able to fit the equilibrium data well, but in most fits, calmodulin was less sensitive to $Ca^{2+}$ than indicated by the data. Some of this behaviour can be attributed to the prior because most of the runs hit the uniform distribution limits (especially the dissociation constants). When the limits were wider, there was a significant amount of training failures (up to 30%) and trained models showed step-like equilibrium behaviour that was overly sensitive to $Ca^{2+}$. Therefore, we opted to keep the narrower limits. Scheme 1 with our parameter sets is generally much more sensitive to $Ca^{2+}$than it is with the rate constants from Kim et al. [9], which result in calmodulin behaviour that does not show appreciable $Ca^{2+}$ binding in the relevant $Ca^{2+}$ range and is significantly right-shifted (Fig 7, top row pink line).

For Scheme 2 (Fig 7 second row from the top), our fits result in behaviour that is significantly more sensitive to $Ca^{2+}$ than the experimental data. Calmodulin would be close to fully bound under resting neuronal $Ca^{2+}$ levels, in contrast to the reaction rate constants from Bhalla and Iyengar [10], which are significantly less sensitive to $Ca^{2+}$ than the data indicate, not reaching full calmodulin saturation in the experimental data range. The failure to fit the equilibrium data is likely due to the inclusion of the dynamical data—reaction rate sets that would allow this scheme to fit equilibrium data do not fit the dynamical data well. The failure of Scheme 2 fitting both dynamical and equilibrium data is likely due to the first $Ca^{2+}$ binding event including two $Ca^{2+}$ ions and needing to be relatively fast to fit the dynamical data.

For Scheme 3 (Fig 7 third row from the top), the parameters from Shifman et al. [11] perform very well because they were explicitly tuned to only this data set. However, when the dynamical data from Faas et al. [5] is included in the fitting procedure, the resulting equilibrium behaviour varies between runs (blue shaded area). Similarly to Scheme 2, our fits result in behaviour that is much more sensitive to $Ca^{2+}$ than the data indicates. However, contrary to Scheme 2, the range of behaviours is much more varied and a significant portion of fits match data from Shifman et al. [11] reasonably well.

For Scheme 4 (Fig 7, fourth row from the top), even though in general the fits are much better, there are still a few runs that do not perform as well. Moreover, our mean RMSE value is slightly worse than that parameters from Pepke et al. [12] for the equilibrium data in Shifman et al. [11]. Curiously the mean RMSE for the dynamics predicted via Scheme 4 with our rate constants is much smaller compared to the rate constants from Pepke et al. [12]. A possible explanation for this is that rate constants from Pepke et al. were first derived using Scheme 5 and then reduced to Scheme 4. Scheme 5 is a more powerful model due to having more state variables which could make optimization easier than simply using Scheme 4 (see results below).

Our fits to the Shifman et al. [11] equilibrium data follow a similar pattern for Schemes 5 and 6. For both Schemes the noisiness in model behaviour that was present for Schemes 1–4 is either gone or significantly smaller, and most fits match the data from Shifman et al. [11] reasonably well. In both cases, the average RMSE value using our rate constants is significantly smaller compared to the published reaction rate constants.

## Model comparison via AIC

We now compare both the published reaction rate constants to our reaction rate constants and between the kinetic schemes via AIC evaluated on the test set of a random seed. AIC is a useful model comparison tool because it takes into account both model predictive performance as well as model complexity (number of parameters). Fig 8 shows the box plots for the AIC values for all the combinations of kinetic scheme and parameter set for all 20 seeds. As shown in Table 2, our reaction rate constants have lower median AIC values (lose less information)

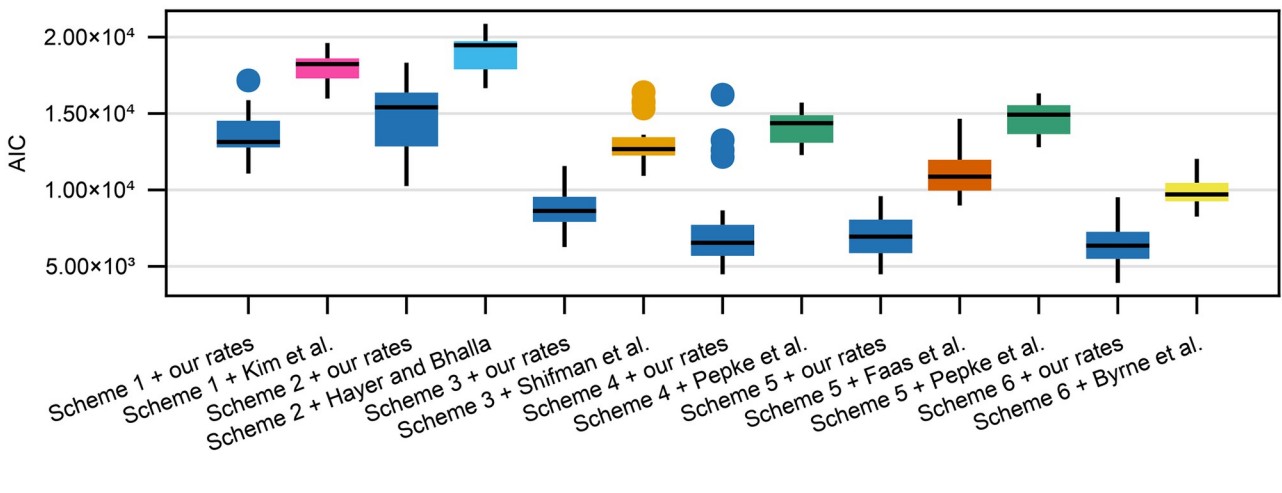

**Fig 8. AIC box plots for all the different combinations of kinetic schemes and either previously published or our own reaction rate constants.**
Each box plot is based on training the model on 20 different random seeds, the AIC value is calculated on the test data set of a given seed.

compared to the published ones. Moreover, the more complex the scheme, the lower the AIC value, with Scheme 6 performing the best. The median AIC values seem to asymptote and reach a lower value by Scheme 6 (the big change is after Schemes 2–3), so only qualitatively different model improvements are likely to decrease the AIC value more.

Calculating the relative likelihoods from median AIC values, where our reaction rate constants are the reference, all the published parameter sets have negligibly low relative likelihoods (largest being on the order of $e^{-100}$). Therefore, our reaction rate constants are significantly more likely compared to the published ones.

Since our reaction rate constants have lower AIC values (lose less information) compared to the published ones, we use our reaction rate constants to compare between different kinetic schemes. Given the results in Fig 8 and Table 2 and using the median AIC for Scheme 6 with our rate constants as reference, the other schemes with our parameter sets have a negligibly small relative likelihoods (again on the order of $e^{-100}$). Therefore, of the combinations of schemes + reaction rate constants that we found, Scheme 6 with our reaction rate constants is relatively the most likely.

## Calmodulin Ca²⁺ integration properties

Having established that our reaction rate constants are significantly more likely than the published ones, we now ask whether this difference is meaningful practically. To answer this

**Table 2. Median AIC values for all combinations of kinetic schemes and reaction rate constants (our fits or published).**

| Parameter source | Median AIC values (×10³) | | | | | |
|---|---|---|---|---|---|---|
| | Scheme 1 | Scheme 2 | Scheme 3 | Scheme 4 | Scheme 5 | Scheme 6 |
| **Ours** | 12.9 | 14.9 | 8.32 | 6.37 | 6.45 | 5.97 |
| **Published** | 17.9 | 18.4 | 11.5 | 14.2 | 10.2[†] / 14.8[‡] | 9.21 |

[†] when reaction rate constants from Faas et al. [5] are used.

[‡] when reaction rate constants from Pepke et al. [12] are used.

question we probe the Ca²⁺ signal integration properties of calmodulin. CA1 pyramidal cell Schaffer collateral synapses undergo long-term potentiation dependent on CaMKII (and therefore on calmodulin) in response to three 1s trains of 50Hz stimulation [39]. Given these results, it is likely that calmodulin integrates the Ca²⁺ signal within a single train. Therefore, we set up a series of simulations where a model was stimulated by a 1s train of Ca²⁺ injections but the frequency was varied from 2Hz to 100Hz. Based on the results in Sabatini et al. [40], Ca²⁺ influx due to single synaptic stimulation event for a neuron at resting voltage is around 0.7µM (this mimics the experimental setup in Bayazitov et al. [39] best). To mimic the competition between calmodulin and other buffers and pumps we implemented a minimal Ca²⁺ extrusion model using values in Sabatini et al. [40] Table 1 for a CA1 pyramidal cell spine— Ca²⁺ decaying to a baseline of 100nM with a time constant $\tau = 12$ms. Finally, we use a biologically realistic calmodulin concentration of 20µM [41]. After the simulation, we evaluate the calmodulin signal integration by calculating the area under the curve of both partially bound calmodulin and fully bound calmodulin, where bigger values indicate a larger level of Ca²⁺ signal integration.

As shown in Fig 9 columns one and two, Schemes 1 and 2 are not capable of integrating Ca²⁺ signals in the tested frequency range (except for a few outlier runs with Scheme 2). For Scheme 1 it is likely due to the fact that the models are not sensitive enough to Ca²⁺ (see Fig 7 top row). The same interpretation, however, does not hold for Scheme 2, whose equilibrium behaviour with our parameter fits was usually too sensitive to Ca²⁺ compared to experimental data. This behaviour for Scheme 2 may be explained by slow reaction rate constants compared to Ca²⁺ decay.

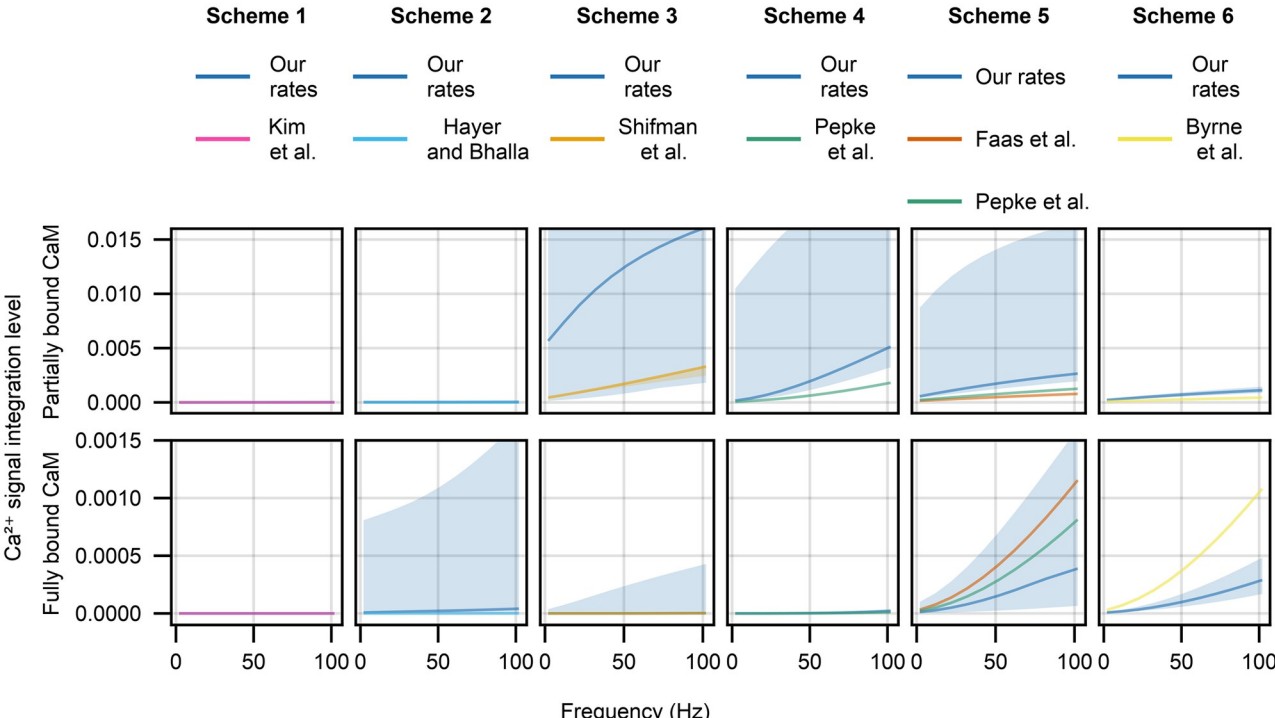

**Fig 9. Ca²⁺ signal integration properties (measured as area under the curve) of partially (first row) and fully bound (second row) calmodulin species in response to a 1sec train of 2–100Hz stimulation that delivers 0.7µM Ca²⁺ per spike (same general patterns hold if 12µM Ca²⁺ per spike is delivered, results not shown).** A different calmodulin scheme is used in each column and shows our parameter fits (deep blue lines) and the published parameter values (all other colours). The solid lines are median model behaviour and shaded areas are the 95% confidence intervals.

The results are somewhat different for Schemes 3 and 4 (Fig 9 columns three and four), where both the published reaction rate constants and our own fits show significant $Ca^{2+}$ integration in the partially bound calmodulin species, but barely any in the fully bound calmodulin. In both cases our reaction rate constants result in significantly higher $Ca^{2+}$ signal integration. However, results for Scheme 3 with our reaction rate constants which show significant $Ca^{2+}$ integration at 2Hz should be taken with caution due to the same fits being overly sensitive to $Ca^{2+}$ at equilibrium (Fig 7 third row from the top).

Finally, for Schemes 5 and 6 we see integration of $Ca^{2+}$ signals that results in both fully and partially bound calmodulin species (Fig 9 columns five and six). For both schemes our reaction rate constants predict that $Ca^{2+}$ signal integration would result in more partially bound calmodulin compared to predcitions from the published rate constants. As for fully bound calmodulin, our reaction rate constants predict a lower level of fully saturated calmodulin than the predcitions from the published rate constants.

The difference between the partially and fully bound calmodulin signals is more pronounced with our reaction rate constants than with the published ones. For Scheme 5 the difference is around an order of magnitude for our reaction rate constants and under 2-fold for the published reaction rate constants, whereas for Scheme 6 the difference is around 4-fold for our reaction rate constants and around 2-fold for the published reaction rate constants. Given that our models reaction rate constants perform better, we predict that partially bound calmodulin species play a more significant part in $Ca^{2+}$ signalling integration and propagation than predicted by previously published models.

## Parameter correlations

We next examine the relationships between our parameter fits within a given scheme. Analysing relationships between parameters may point future experimental research questions. For example, if some reaction rate constants are correlated, they may be under-determined. Therefore, future model development would benefit from additional, more directed data to better constrain the correlated parameters. We use partial correlation as a measure of relationship between parameters [19]. Briefly, partial correlation quantifies the degree of association between two variables when the variance from a set of controlling variables is taken into account. For example, in Scheme 1 the partial correlation between $k_1$ and $k_3$ would indicate the relationship between these two rate constants when $K_{D_1}$ and $K_{D_1}$ is accounted for.

Since structurally there is nothing to distinguish between the C and the N lobes for Schemes 4–6, we calculate the dissociation constant for the two binding reactions in Scheme 4 and for the first binding reaction for both lobes in Scheme 5 and Scheme 6 and compare their values—the one that has a lower $K_D$ value we call the C lobe and the one that has a higher value we call the N lobe. This is to avoid what we call the C lobe being functionally the N lobe and vice versa, which would result in artificially higher parameter spread or obscure parameter correlations. We show the parameter pair plots for all six schemes in the S5 Appendix, along with tables of individual reaction rate fits. We show partial correlation coefficients for all schemes in Fig 10.

First of all, even though the data set we use is richer, as it includes both the dynamical and the equilibrium data, there is still significant variance in our model parameter fits. For some parameters the pairs can span 5 to 10 orders of magnitude (see S5 Appendix). Moreover, there are significant correlations between multiple parameters in most schemes.

There is only one significant negative correlation for Scheme 1, between $K_{D_1}$ and $K_{D_2}$. This correlation is most likely due to a limited number of degrees of freedom offered by this scheme. Assuming that a calmodulin molecule has an overall dissociation constant that is a

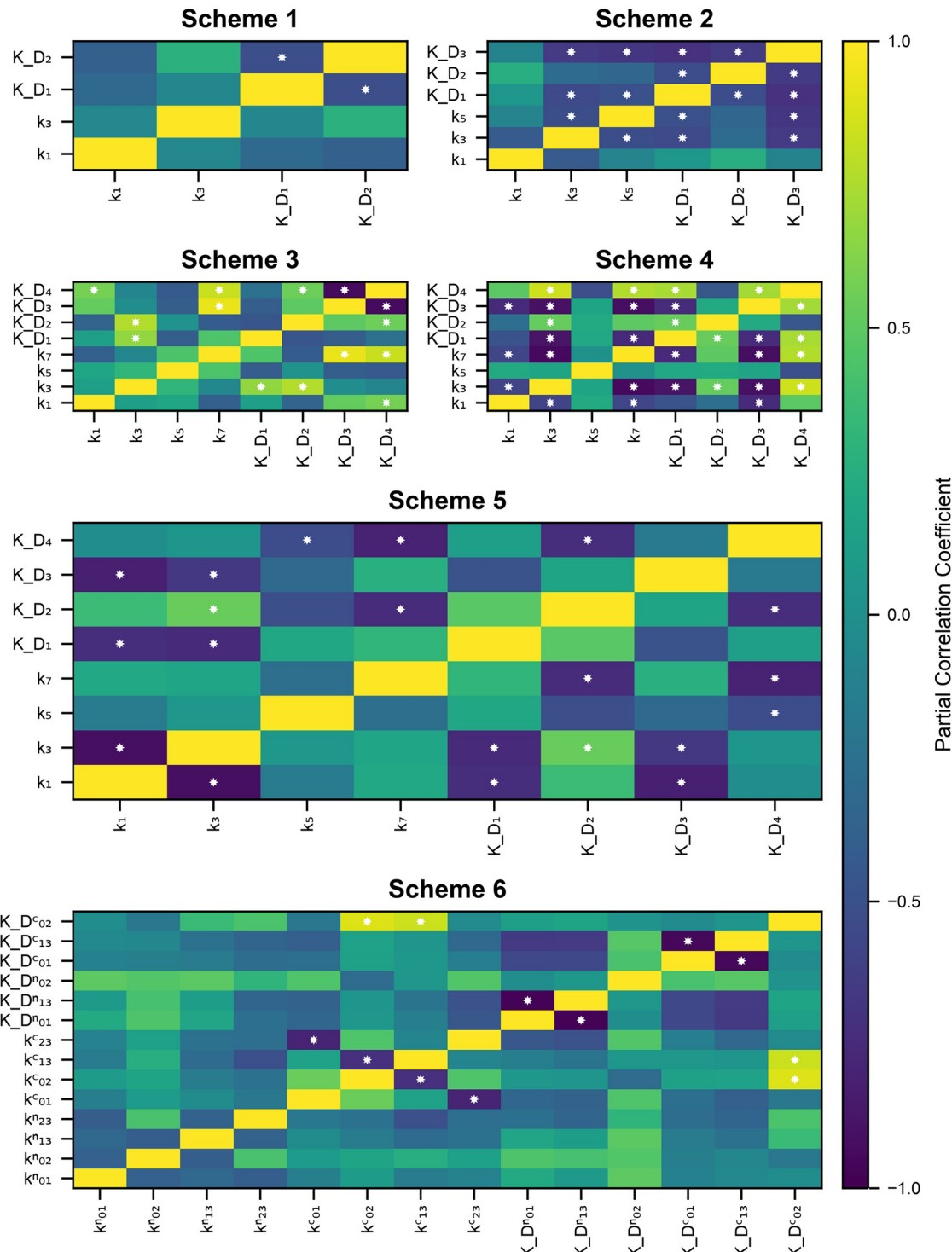

**Fig 10. Partial correlation coefficients for our reaction rate constant fits for all schemes.** White stars in the rectangle indicate $p < 0.05$ with null hypothesis that the partial correlation is zero.

function of the dissociation constants of individual reactions, the dissociation constants of individual reactions have to co-vary in order to maintain the same overall behaviour.

Reaction rate constants for Schemes 2 have 8/15 significant correlations which indicates a high level of sloppiness in the system. Similarly to Scheme 1, given the high level of simplification used in this scheme, in order to maintain the same overall model behaviour, parameters have to co-vary. This is especially true for the dissociation constants, which are all are negatively correlated. We also see that the on-rate for the first reaction is not correlated to any other parameter and is therefore fairly well-determined.

Scheme 3 has proportionally fewer correlations than Scheme 2 (7 out of 28 parameters correlated). Most correlations are with the dissociation constant for the final $Ca^{2+}$ binding event. This can be explained by thinking about a general dissociation constant for calmodulin, which would be a function of $K_{D_1}$, $K_{D_2}$, $K_{D_3}$ and $K_{D_4}$—since $K_{D_1}$ is likely better constrained due to the first binding event needing to be of particular speed to fit the rising/falling phase of the individual time series, the other dissociation constants may co-vary more freely and balance each other out. The on-rate for the first binding event is only correlated to one variable, which indicates that it is quite well determined.

Scheme 4 shows the largest fraction of correlations of all the schemes (15/28). This is most likely due to the quasi-steady state approximation which results in steady state reaction rate constants $\{k_i^{ss}\}_{i=1}^4$ that provide a lot of room for sloppiness via products and quotients of the full set of reaction rate constants $\{k_i\}_{i=1}^8$.

The point that model structure is of utmost importance in determining the levels of sloppiness in the system is further reinforced by Scheme 5, where 10 out of 28 reaction rate constants were correlated. More importantly, a significant number of correlations are within-lobe, for example $k_1$ and $k_3$—the first and second on-rate constants. There are also some cross-lobe correlations, for example $K_{D_2}$ and $K_{D_4}$ which are the second $Ca^{2+}$ binding events for C and N lobes respectively.

Curiously, even though Scheme 6 is the most complex in terms of number of parameters and number of states, it shows only six significant correlations between reaction rate constants. Moreover, all correlations are within a lobe, rather than between lobes. More specifically, most of them are for parameters in the C lobe, rather than the N lobe.

## Necessary structural components of a calmodulin model

As shown in Table 1 and in Fig 6, there is a large gap in training performance between Schemes 1–4 and Schemes 5–6. Even though training RMSE in both dynamical and equilibrium data significantly decreases going from Scheme 2 to Scheme 3, only from Scheme 5 onwards can both dynamics and equilibrium behaviour be captured well. There are two main differences between Schemes 1,2,4 and and 5–6: independence of lobes and structural assumption of co-operativity. Both Scheme 3 and Schemes 5–6 allow co-operativity (via reaction rate constants) but do not assume it structurally. Schemes 3 does not allow for independence of lobes, while Schemes 5–6 assume it structurally. In this section we provide an empirical argument that links model features to gaps in performance, focusing on event-based (as opposed to binding site-based) and structurally co-operative (especially for the C lobe) schemes to model calmodulin.

Assuming that the real calmodulin dynamics operate in a $k$-dimensional space, any model capable of modeling the dynamics would have to have at least that many dimensions (along with an appropriate structure). Calmodulin models framed in terms of events (fully abstracted from binding sites) can operate at most in a four dimensional linear subspace (since rank of such a network is four, see page 30 in [42]) of the five dimensional state space (see Scheme 3 in

Fig 3). Therefore, an immediate conclusion of this may be that $k > 4$, real calmodulin dynamics operate in a higher dimensional space than an event-based model allows for. However, Scheme 5, which is able to model both calmodulin dynamics and equilibrium behaviour (see Table 1), has rank 4 as well. The main difference between Schemes 3 and 5 are the independence of the lobes: Scheme 5 contains two independent subnetworks (each of which is rank 2). Therefore, based on our results, in order to accurately model both calmodulin dynamics and equilibrium behaviour, two independent subnetworks (independence of lobes) is a necessary model feature.

We next analyze whether a structural assumption of co-operativity, modelling the binding of two $Ca^{2+}$ ions as a single event, within calmodulin lobes is reasonable. This is not the only way of modelling co-operativity, but it results in models with a smaller state space vector and therefore can be preferable computationally. Fractional calmodulin occupancy of the N and the C lobes using a well performing model (Scheme 6 with parameters from Byrne et al. [13]) is shown in Fig 11 columns one and two. Starting with the dynamics of the partially occupied N lobe, the model predicts around 20% of calmodulin molecules would have the first site occupied, with a negligible fraction having the second site occupied. Moreover, the dynamics of partially occupied sites in the N lobe do not show fast changes over the simulated time period, so the quasi-steady state approximation would hold reasonably well. The dynamics of the C lobe paint an opposite picture. It is immediately obvious that, due to its slower speed, the quasi-steady state approximation ($d[CaMC_1]/dt = 0$) does not hold for the C lobe as there are calmodulin dynamics occurring over the whole simulated time of 35ms. Therefore, even though it is a theoretically appealing tool to reduce the number of calmodulin states, the quasi-steady state approximation is too inaccurate for the C lobe and results in significant errors in either calmodulin dynamics or equilibrium behaviour.

## Discussion

We used a rich dynamical [5] and equilibrium [11] data set to fit six calcium-calmodulin kinetic schemes from scratch in order to compare to published models. Our comparison resulted in a number of conclusions. First of all, the parameters we found, as opposed to the published ones, resulted in significantly better fits on our dataset (Table 1). Secondly, we showed that fully event-based schemes that do not utilize any features of the calmodulin physical structure (existence of C and N lobes) result in significantly worse generalization performance as measured via AIC (Fig 8). Thirdly, we investigated calmodulin signal integration properties by comparing our parameter fits to published reaction rate constants for different calcium-calmodulin schemes. Some schemes showed no $Ca^{2+}$ signal integration in response to a stimulation protocol mimicking an empirically effective plasticity induction protocol highlighting the importance using more detailed calmodulin schemes (Fig 9). Fourthly, we calculated the partial correlations between our parameter fits (Fig 10). Partial correlations revealed that even with our data set, that is richer than anything used before, some parameters were correlated and therefore under-determined. Finally, we investigated the validity of the quasi-steady state approximation used in [12] and by using Faas et al. [5] data we showed that it is not accurate for the C lobe. We next discuss each of these conclusions individually.

First of all, model performance depends on the data which was used to parameterise it. Even though usage of multiple data sources to fit a calmodulin model is not new and was done in Pepke et al. [12], we are the first to combine a data source on calmodulin dynamics [5] and a data source on calmodulin equilibrium behaviour [11]. We used this combined data set to fit six different calcium-calmodulin kinetic schemes previously used in the literature. We then compared our parameters to the published ones which revealed that a significant number of

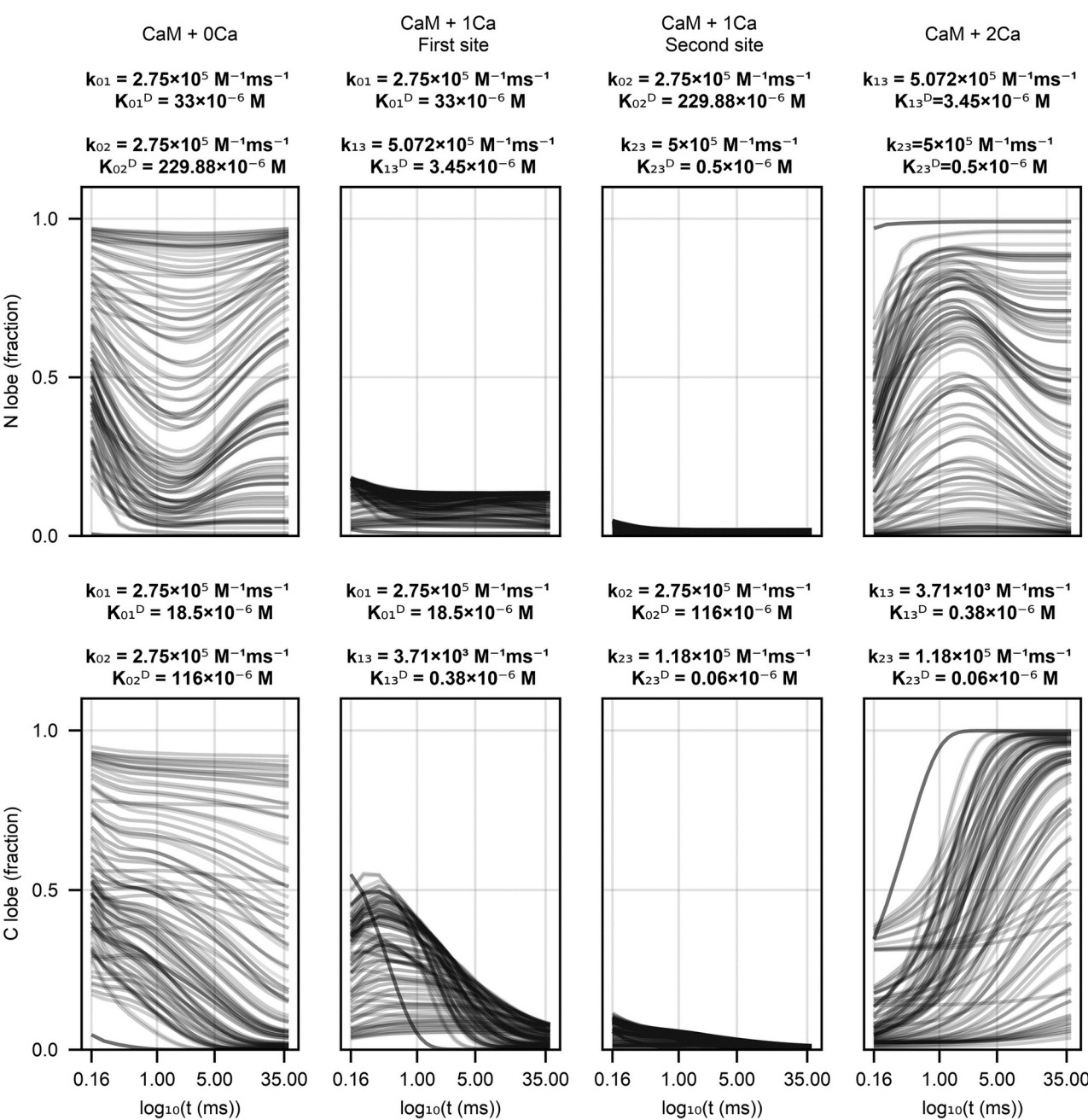

**Fig 11. Scheme 6 behaviour with parameters from Byrne et al. [13] on dynamical data from Faas et al. [5].** Each line is one of the initial conditions (solution + uncaging strenght) Faas et al. used. Calmodulin has been normalized to total calmodulin used in an experiment, the first row shows N lobe dynamics, the second row shows C lobe dynamics. Each column shows a different calmodulin state—completely unbound (first column), $Ca^{2+}$ bound to the first site on a lobe (second column), $Ca^{2+}$ bound to the second site on a lobe (third column), $Ca^{2+}$ bound to both sites of a lobe (fourth column). Above the plots we provide reaction rate constants from Byrne et al. [13] for the reactions a specific calmodulin state participates in. Note that time on the x axis is in log10 space to better show the initial dynamics.

calcium-calmodulin models used in the literature are parameterized sub-optimally (see Table 1). Most published models (except Faas et al. [5]) have relied on either only equilibrium data [11, 12, 43] or dynamical data obtained under significant methodological limitations—such as dead time in stopped flow fluorimetry or presence of other biochemical species [9, 13]. Undoubtedly, it would be unfair to criticize past work for operating under the limitations of the day, but that does not prevent models from becoming outdated (much as this work will be one day). Therefore, an important contribution in this paper are the improved model parameters for calcium-calmodulin models—the best performing parameter sets for each scheme are given in Table 3 (see S5 Appendix for all 20 parameter sets for each scheme).

Secondly, our calmodulin model comparison uncovered discrepancies in performance between different kinetic schemes. The complexity of calmodulin schemes we investigated ranged from a model with three states and four parameters [9] to a model with eight states and sixteen parameters [13]. There were only two schemes (5 and 6, consisting of eight and sixteen parameters respectively) that were able to fit both sources of data well—both schemes modelled calmodulin lobes separately and consisted of individual, rather than lumped, $Ca^{2+}$ binding reactions. Two further schemes (3 and 4), one of which modelled calmodulin lobes but not

**Table 3. Our reaction rate sets that performed best on the test data and the published reaction rate constants from literature.** All parameters are in $\log_{10}$, but are in different units, depending on the context: for second order reactions the forward reaction rate constants are in $M^{-1}ms^{-1}$, dissociation constants in M, for third order reactions the forward reaction rate constants are in $M^{-2}ms^{-1}$, dissociation constants in $M^2$.

| Source | Parameters | | | | | | | | | | | | | | |
|---|---|---|---|---|---|---|---|---|---|---|---|---|---|---|---|
| | **Scheme 1 parameters** | | | | | | | | | | | | | | |
| | $k_1$ | | $K_{D_1}$ | | | $k_3$ | | | $K_{D_2}$ | | | | | | |
| Our fits | 8.10 | | -9.00 | | | 4.00 | | | -9.00 | | | | | | |
| Kim et al. | 3.60 | | -5.69 | | | 5.0 | | | -4.96 | | | | | | |
| | **Scheme 2 parameters** | | | | | | | | | | | | | | |
| | $k_1$ | | $K_{D_1}$ | | $k_3$ | | $K_{D_2}$ | | | $k_5$ | | $K_{D_3}$ | | | |
| Our fits | 8.13 | | -9.00 | | 2.25 | | -9.00 | | | 4.52 | | -6.40 | | | |
| Hayer and Bhalla | 4.86 | | -6.00 | | 3.56 | | -5.55 | | | 2.67 | | -4.68 | | | |
| | **Scheme 3 parameters** | | | | | | | | | | | | | | |
| | $k_1$ | $K_{D_1}$ | | $k_3$ | | $K_{D_2}$ | | $k_5$ | | $K_{D_3}$ | | $k_7$ | | $K_{D_4}$ | |
| Our fits | 5.41 | -4.87 | | 2.36 | | -7.83 | | 4.33 | | -6.40 | | 5.04 | | -6.19 | |
| Shifman et al. | - | -5.10 | | - | | -5.77 | | - | | -4.46 | | - | | -5.05 | |
| | **Scheme 4 parameters** | | | | | | | | | | | | | | |
| | $k_1$ | $K_{D_1}$ | | $k_3$ | | $K_{D_2}$ | | $k_5$ | | $K_{D_3}$ | | $k_7$ | | $K_{D_4}$ | |
| Our fits (S4) | 4.33 | -5.56 | | 5.98 | | -6.01 | | 7.03 | | -4.13 | | 6.29 | | -5.73 | |
| Pepke et al. | 3.60 | -5.00 | | 4.00 | | -6.03 | | 5.0 | | -4.60 | | 5.18 | | -5.30 | |
| | **Scheme 5 parameters** | | | | | | | | | | | | | | |
| | $k_1$ | $K_{D_1}$ | | $k_3$ | | $K_{D_2}$ | | $k_5$ | | $K_{D_3}$ | | $k_7$ | | $K_{D_4}$ | |
| Our fits | 4.50 | -5.43 | | 4.08 | | -5.84 | | 5.69 | | -4.00 | | 6.32 | | -5.30 | |
| Faas et al. | 4.90 | -4.60 | | 4.40 | | -6.60 | | 5.90 | | -3.70 | | 7.50 | | -6.10 | |
| | **Scheme 6 parameters** | | | | | | | | | | | | | | |
| | $k_1^c$ | $k_3^c$ | $k_5^c$ | $k_7^c$ | $K_{D_1}^c$ | $K_{D_2}^c$ | $K_{D_3}^c$ | $K_{D_4}^c$ | $k_1^n$ | $k_3^n$ | $k_5^n$ | $k_7^n$ | $K_{D_1}^n$ | $K_{D_2}^n$ | $K_{D_3}^n$ | $K_{D_4}^n$ |
| Our fits | 4.23 | 4.16 | 5.22 | 2.31 | -5.40 | -4.38 | -5.38 | -6.40 | 3.03 | 6.65 | 5.22 | 7.58 | -4.06 | -5.20 | -2.88 | -6.38 |
| Byrne et al. | 5.44 | 5.44 | 3.57 | 5.07 | -4.73 | -3.94 | -6.42 | -7.22 | 5.44 | 5.44 | 5.71 | 5.70 | -4.48 | -5.46 | -3.64 | -6.31 |

Shifman et al. only contained the dissociation constants, so the forward reaction rate constants have no point of comparison. The same set of reaction rate constants from Pepke et al. has been used in both Schemes 4 and 5, but they are only shown for Scheme 4 to avoid repetition and misleading as the implementation of Scheme 5 in Faas et al. and Pepke et al. is slightly different, structurally $k_{Pepke} = 2k_{Faas}$ for some reaction rate constants.

individual binding, another which modelled individual binding but not lobes, were able to fit dynamical data, but not equilibrium data, reasonably well. Both of these schemes consisted of eight parameters, same as one of the schemes that fit both sources of data well, indicating that the number of parameters is not the only factor necessary for an accurate calcium-calmodulin model. Finally, two of the simplest schemes (Schemes 1 and 2) that did not model calmodulin lobes and modelled $Ca^{2+}$ binding as lumped reactions were not able to fit either the dynamical data or the equilibrium data well. These results, along with median AIC values (Table 2) lead to the second contribution of this paper—Scheme 6 is the most accurate calcium-calmoduling binding scheme and, compared to some simpler schemes, by a significant margin.

Thirdly, our results provide implications for models that include calmodulin. We investigated the $Ca^{2+}$ integration properties of calmodulin in response to a realistic $Ca^{2+}$ spike train (see Fig 9). The biggest practical difference between our reaction rate constants and published ones is that there is a much more significant contribution from partially bound calmodulin species, rather than fully bound calmodulin. As shown in Shifman et al. [11], CaMKII can be activated by partially bound calmodulin. Moreover, calmodulin has many binding partners, such as Calcineurin [44], Phosphodiesterase 1 [45], Adenylyl cyclases 1 and 8 [46], Neurogranin [47, 48] and others [2]. Our results bring into question the accuracy of the results of publications where poorer performing schemes or parameterisations are used in larger models [9, 18, 19, 21–23, 49–52]. There are many ways to compensate for the poor performance of calmodulin scheme or parameters. For example, it is possible that in some cases the lack of calmodulin sensitivity to $Ca^{2+}$ has been compensated for by an increased $Ca^{2+}$ influx. However, for example Scheme 1 is used in [23] in a dynamical setting, stimulating their large model with many protein species with e. g. 180s of 5Hz or 1sec of 100Hz $Ca^{2+}$ pulses. As our results show, the calmodulin $Ca^{2+}$ integration properties are significantly different in this range when our reaction rate constants are used. Our third contribution is support to the hypothesis that partially bound calmodulin molecules arising in response to different $Ca^{2+}$ stimuli is an additional dimension of signal encoding and propagation towards downstream pathways compared to spatial/concentration based fully bound calmodulin signalling. Future investigations into other calmodulin binding partners and their activation by partially bound calmodulin species would be able to falsify this hypothesis.

Fourthly, our results on the partial correlations between reaction rate constants form our fourth contribution—the call for more empirical investigations to test the distinctness of $Ca^{2+}$ binding sites within a calmodulin lobe. Generally with increasing model complexity there were fewer correlations (except for Scheme 4, which had more than Scheme 3) between parameters, indicating the parameters were better determined by data. However, even for the most complex Scheme 6, there were some correlations between parameters within the same lobe. These correlations could only be eliminated by additional information on the properties of individual binding sites. Existing studies with mutations of individual calmodulin binding sites only include equilibrium measurements [11, 53, 54]. Since equilibrium behaviour only informs the ratio between the $Ca^{2+}$ binding and unbinding rate constants, they are of limited usefulness in fitting. The closest to the necessary measurements were done in Faas et al. [5] where dynamical measurements with one inactive calmodulin lobe (either C or N) were made.

Finally, we investigated the validity of the quasi-steady state approximation used in [12]. Both Scheme 4, in which partially bound calmodulin species are not modelled due to the quasi-steady state approximation, and Scheme 5, which models them, can model calmodulin dynamics to a similar accuracy. The main difference between the schemes is in equilibrium behaviour, where in Scheme 4 the modelling of dynamics impedes modelling of steady state behaviour. These results imply that the quasi-steady state approximation used in Pepke et al. [12] does not hold in the context of the Faas et al. [5] data, at least not without significant

decrease in the accuracy of model behaviour. Ideally, an empirical measurement of the occupancy of individual calmodulin sites in a dynamical setting would be definitive in falsifying this approximation. Unfortunately, such data does not exist therefore we used Scheme 6 with Byrne et al. [13] parameters (since they fit the data reasonably well) and simulated the fractional occupancy of individual calmodulin sites under [5] experimental conditions (see Fig 11). These results support our fifth contribution—that the quasi-steady state approximation is not valid and results in a significant loss of accuracy, especially for the C lobe.

Having discussed the contributions of this paper we now reflect on their wider implications and practical reality of computational modelling. Suboptimal schemes or parameterisations of calcium-calmodulin models used in large models are a difficult challenge. It is not necessarily the case that the conclusions drawn from large models are made invalid. In large models it is likely possible to correct for the model-data mismatch arising due to inaccurate calmodulin behaviour via the parameters of reactions involving downstream molecules. This, however, may result in a panoply of different mechanistic hypotheses if different publications correct for these inaccuracies arising due to poor calmodulin models in different ways. A more co-ordinated community effort with some agreed upon set of model tests (such as the FindSim platform suggested by [55]), akin to continuous integration in GitHub, may be necessary to resolve such issues in the future and build performant large models.

Limited computational resources and the difficulty of writing large models mean that in some cases it may not be feasible to use a more detailed calmodulin scheme because of an exponential explosion in the number of species to be modelled and the subsequent increase in the computational cost of simulations. Rule-based modelling [56] with its "don't care, don't write" approach (only having to specify the features of a species which impact a reaction) allows models containing exponentially large numbers of complexes to be written down but may still be too computationally costly. Modeling is a complex task that involves many behind the scenes choices about acceptable trade-offs. Our results provide the information about the trade-offs in model accuracy being made when choosing one calmodulin scheme (or parameter set) over another.

In the final two paragraphs we discuss the methodology we used, the available alternatives and limitations. We used NLME fitting algorithms implemented in `Pumas.jl` to fit the reaction rate constants of the different calcium-calmodulin kinetic schemes. There are many published pipelines for fitting reaction rate constants of kinetic schemes. For example, Eriksson et al. [57] propose and use a pipeline based on approximate Bayesian computation Markov Chain Monte Carlo (ABC-MCMC, using R-vines). MCMC approaches are powerful tools which benefit from inherently providing uncertainty on model parameters, rather than having to run optimization on different random seeds as was done in this study. However, they are generally much more computationally expensive. Another popular option is the Data2Dynamics toolbox [58], which streamlines construction of models of chemical reaction networks and modeling of experiments while leveraging ODE solving capabilities of MATLAB, along with stochastic optimization. However, there are few modern software packages that deal with NLME models (which were required due to the nature of the dynamical data in Faas et al. [5]). Of these packages `Pumas.jl` is currently the most performant one [24]. This is in part because `Pumas.jl` is implemented in the Julia programming language which contains state of the art ODE solving capabilities, outperforming its competitors in terms of speed by orders of magnitude (see benchmarks.sciml.ai).

Even with a powerful computational pipeline, there are still many nuances, practical considerations and limitations. For example, the length of the time series to which parameters are being fit impacts the complexity of the loss surface—the more points, the more complex it is [59]. Therefore, we downsampled the initial part of the dynamical data from Faas et al. [5] (see S1 Fig). However, invariably, downsampling results in loss of signal, therefore more

performant downsampling techniques or multiple shooting based approaches may have resulted in even better fits. Moreover, we simplified the $Ca^{2+}$ uncaging model used in [5] to make parameter optimization more stable. Also, [5] used Pockels cell delay (PCD) as the independent variable to predict the fraction of uncaged $Ca^{2+}$ whereas we omitted this variable as it did not perform as well in practice. More data on the relationship between PCD and $Ca^{2+}$ uncaging fraction would have allowed us to derive a better $Ca^{2+}$ uncaging model that potentially could have improved model predictions with both published and our own reaction rate constants. Finally, in order to prevent training failures due to numerical instabilities in ODE solutions when using some schemes, we had to restrict the range of possible values taken by their reaction rate constants. Usage of novel ODE solvers capable of handling stiff systems is a potential avenue to remedy this limitation in future studies. Therefore, even with a more powerful software pipeline, some trial and error and practical trade-offs were necessary to fit our own parameters and efficiently and accurately compare different calmodulin models.

In conclusion, we believe that we have provided a number of important contributions that advance calcium-calmodulin modelling. We conducted a data-driven evaluation of both calcium-calmodulin kinetic schemes and parameter sets used in existing publications and showed which schemes or parameter sets performed poorly. It may be argued that behaviour of single molecules in large models matters less than the behaviour of the overall model. However, if large models are to be useful in predicting the behaviour of real biological systems, the individual molecules and their accurate generalization performance are of utmost importance.

## Supporting information

**S1 Appendix. Subset of data used and its splitting into training, validation and testing data sets.**
(PDF)

**S2 Appendix. Concentrations of species of different groups of solutions used in Faas et al. data.**
(PDF)

**S3 Appendix. Derivation of steady-state reaction rate constants for Scheme 4.**
(PDF)

**S4 Appendix. Published reaction rate constants used in this study.**
(PDF)

**S5 Appendix. Our parameter fits for all schemes.**
(PDF)

**S1 Fig. Comparison of the original data set and the data with subsampled initial period.**
(TIF)

## Author Contributions

**Conceptualization:** Domas Linkevicius, Melanie I. Stefan, David C. Sterratt.

**Data curation:** Guido C. Faas.

**Investigation:** Domas Linkevicius.

**Methodology:** Domas Linkevicius, Melanie I. Stefan, David C. Sterratt.

**Software:** Domas Linkevicius.

**Supervision:** Angus Chadwick, Melanie I. Stefan, David C. Sterratt.

**Visualization:** Domas Linkevicius, Melanie I. Stefan, David C. Sterratt.

**Writing – original draft:** Domas Linkevicius.

**Writing – review & editing:** Domas Linkevicius, Angus Chadwick, Guido C. Faas, Melanie I. Stefan, David C. Sterratt.

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
