## [Decision Letter · Decision Letter 0]

29 Nov 2024

PONE-D-24-49899Fitting and comparison of calcium-calmodulin kinetic schemes to a common data set using non-linear mixed effects modellingPLOS ONE

Dear Dr. Linkevicius,

Thank you for submitting your manuscript to PLOS ONE. After careful consideration, we feel that it has merit but does not fully meet PLOS ONE’s publication criteria as it currently stands. Therefore, we invite you to submit a revised version of the manuscript that addresses the points raised during the review process.

We look forward to receiving your revised manuscript.

Kind regards,

Pan Li, PhD

Academic Editor

PLOS ONE

Journal Requirements:

Reviewers' comments:

Reviewer's Responses to Questions

**Comments to the Author**

1. Is the manuscript technically sound, and do the data support the conclusions?

Reviewer #1: Yes

Reviewer #2: Yes

2. Has the statistical analysis been performed appropriately and rigorously? 

Reviewer #1: Yes

Reviewer #2: I Don't Know

3. Have the authors made all data underlying the findings in their manuscript fully available?

Reviewer #1: Yes

Reviewer #2: Yes

4. Is the manuscript presented in an intelligible fashion and written in standard English?

Reviewer #1: Yes

Reviewer #2: Yes

5. Review Comments to the Author

Reviewer #1: The main contribution of the manuscript is fitting a set of calcium-calmodulin kinetic models to data from two different sources viz., dynamical/transient data from Faas et.al. [5] and data of equilibrium states from Shifman et. al. [11]. These models are borrowed from the literature and these correspond to varying levels of complexity in their Ca2+ binding schemes. In my opinion, the manuscript lacks novelty, since, there is no new modelling scheme that the authors propose, nor do they propose novel schemes for data fitting. All the models as well as data fitting schemes used in the manuscript are known and the only contribution of the authors is the use of data from two different sources in order to fit parameters that match both data sets using well known methods. The other contribution is an analysis of the six different model schemes in the light of the new parameters that they have estimated. Since there is no novelty in terms of a modelling or estimation/data fitting approach or an experimentally discovered unknown biological phenomenon, I believe that the manuscript is not eligible to be published in PLOS One. In summary, there is neither a mathematical/computational novelty nor a biological/experimental novelty in the manuscript.

Another general remark that I have is the lack of explanation of the parameter estimation procedure. Although the authors have mentioned exactly the known packages/programming languages that have been used, the authors could put more efforts in explaining in detail the underlying mechanism for estimating the parameters. I have a very specific remark regarding the Methods section in page 9. It is not clear what F denotes in equation (7).

Some other remarks are as follows:

1. In page 6, 7, one of the references does not appear, instead it is replaced with ??, see e.g. the end of line 3 in page 7 and the caption of Fig. 3 in page 6.

2. The second paragraph of page 7 needs more explanation. Why the nine states described in the beginning of the paragraph can be reduced to 5 states? What is the biological implication of this reduction?

3. At several places, the term "rates" actually means "rate constants". For example, the term "rates" has been used to mean "rate constants" in Table 1. By the way, no rows have been highlighted in this Table as claimed at the beginning of its caption.

4. There is a typo in line 3 of page 10. Correct the spelling of "available".

5. You have mentioned in page 12 as well as in the Discussion section that the parameters that you have estimated fit the data of Faas et.al. [5] and Shifman et. al. [11] better than the parameters in existing literature in many cases as also seen from the RMSE values in Table 1. I think this is obvious, because the published rate constants may have been derived using other sets of data.

6. In page 16, 17, there is a talk of frequency, periodic signal trains etc. How is this incorporated in your model and how did you simulate these conditions? The model that is currently described in equation (5) does not depend on any of these characteristics (frequency/signal trains etc.).

Reviewer #2: Terminology

Line 239: The authors use the terminology of Rackaukas et al. (2022) [1] and call the function g the “structural model”. I find this to be misleading, if not plain wrong. Please compare e.g. the terminology of Duchesne et al. (2021) who use definitions that I find correct: The structural model describes the dynamics of the system (could be an ODE model as described by u’ in Equation 4), the function g would then be the parameter model, and then there is an error model (here a simple observation noise standard deviation).

Also, I think that using the symbol eta as a parameter in equation 2 is a suboptimal choice, because in my mind, eta is the deviation of a random effect parameter from the typical value, i.e. should have a mean of zero (also compare [2]). Correspondingly, only the random effect covariance Omega enters the random effects in Figure 4, while the eta in equation 2 is also influenced by mu, which would be part of the fixed effects parameter vector theta.

The use of eta for the uncaging model led me to believe that a random effect was only assumed for the uncaging of calcium, and not for all the individual rate constants. Please clarify early in the manuscript where random effects were employed. It seems to me, that calmodulin molecules should be sufficiently homogeneous that no random effects are needed to account for “interindividual differences”, but I am not sure from the model description if random effects were incorporated for the k values.

Optimization method

The explanation to equation (8) suggests that the conditional likelihood was optimized. If I read this correctly, the covariance matrix Omega was fixed in the optimizations? If this is the case, then the formulation “requires appropriate constraints on Omega” (line 266) is misleading. Please clarify.

Model comparison

Only the AIC is explained in the Methods section, but in Table 1, the AIC values are not shown but a comparison of RMSE values by using Cohens d. Please clarify in the methods section how the model comparison was set up (20 model fits), how Cohens d was calculated and if it is appropriate (is the number of model parameters equal in the comparisons?).

This leads me to the AIC – please document how the number of parameters is counted. This is not self-understood in the mixed modelling community, as far as I know different versions exist.

If you use the conditional likelihood in the AIC omputation I think this means that you actually have to count all individual random effects eta_n to have a valid comparison. Please specify the method of likelihood computation used for the AIC values in Table 2. You mention it is easy to calculate (line 315), but e.g. the saemix R package provides three different methods for the likelihood computation, because the full likelihood does not have closed form and has to be approximated, e.g. by Gaussian quadrature, importance sampling or a linearization.

“Model comparison via AIC”

Reading this section (line 426ff) makes me wonder if the increasing complexity of the models is penalized properly, as the observed variables are quite simple and do not allow to distinguish between all the different states of calmodulin. But maybe I am too suspicious and your model specification and fitting method is just very efficient.

Parameter correlations

In NLME models, if a parameter correlation is suspected, this can be specified in the random effects covariance matrix Omega. Now I still do not know if random effects were allowed for the k values. By now this should be very clear to the reader (but I cannot exclude that it is just me not being able to comprehend, please excuse me if this is the case).

Various further notes:

The acronym NLME should be introduced at its first occurrence.

p. 6/31 One reference is not correctly specified in the legend to Figure 3 (“??”). Presumably the same reference concerning Scheme 4 is also not resolved on page 7/31.

p. 7 Please specify the reference by author and year, in addition to the number on line 200

p. 7 Parameter eta in equation 1: is that a random variable? What distribution is assumed, if yes?

The fact that Equation 2 fits better than Equation 1 means that the strength of the laser pulse is completely random and cannot be controlled by specifying a PCD to a relevant degree. I think this should be clarified.

Line 233: “The basis of NLME”? Please clarify. Maybe “The basic NLME”?

Line 605: Closing parenthesis missing

Line 655: Spike train instead of spike trains?

References: Please fix ref 11.

6. PLOS authors have the option to publish the peer review history of their article (what does this mean?). If published, this will include your full peer review and any attached files.

Reviewer #1: No

Reviewer #2: No

---

## [Author Response · Author response to Decision Letter 0]

7 Jan 2025

We are grateful to the Reviewers for their insightful comments and observations regarding our manuscript. We will first address what we see as the major points of discussion, followed by more minor points.

Reviewer #1 wrote: “In my opinion, the manuscript lacks novelty, since, there is no new modelling scheme that the authors propose, nor do they propose novel schemes for data fitting.”

It is true that the current manuscript does not contain a novel calmodulin scheme or a new fitting technique. However, creation of novel models or fitting techniques is not the only possible source of novel scientific insight. The calmodulin model comparison work contained in this manuscript has not been done before and, therefore, leads to new insights: the needlessness of new calmodulin modeling schemes given the current available data, the level of detail necessary to model calcium-calmodulin binding. Furthermore, the work highlights the potential pitfalls of using calmodulin schemes that are too simple to model calcium-calmodulin binding, which calls for a reassessment of existing work. Therefore, despite the manuscript not containing new modeling schemes or parameter estimation methods, it is still original and insightful.

Reviewer #1 also wrote: “You have mentioned in page 12 as well as in the Discussion section that the parameters that you have estimated fit the data of Faas et.al. [5] and Shifman et. al. [11] better than the parameters in existing literature in many cases as also seen from the RMSE values in Table 1. I think this is obvious, because the published rate constants may have been derived using other sets of data.”

The fact that our estimates result in better fits than the previously published rate constants is obvious only at a first glance. Upon deeper reflection it brings up a number of important points, especially taking into account the magnitude of the difference in some comparisons. Even though different published rate constants have been derived using different sets of data, there still ought to be broad agreement in model predictions if the data sets are broadly consistent. If the model predictions are not consistent (especially qualitatively, rather than quantitatively), then either the different data sets are contradictory or some of the models are not powerful enough to fit certain data sets (or both). Our comparisons revealed a number of cases where the differences are qualitative, rather than quantitative, namely Schemes 1–3 and parameters from Pepke et al. show significant qualitative differences from our fits. This, however, is not the case for the Faas et al. or the Byrne et al. parameters, where the differences are much smaller and keep to the same qualitative trends as our fits. Therefore, yes, it is unsurprising that our fits result in lower errors on the data we use to fit them than other models which were tuned to different data sets,

but the size of errors is, in some cases, surprising.

Reviewer #1 also pointed out that the parameter estimation procedure could be explained better. We have added additional explanations, expanding and deepening the level of explanation. We have also addressed the following list of more minor remarks (following the numbering in the review):

1. In page 6, 7, one of the references did not appear; this has been corrected.

2. The explanation of why the nine states described in the beginning of the second paragraph of page 7 can be reduced to 5 states was found to be lacking. We clarified this point and

used more precise language, making it clear that this is not a conceptual reduction of the model, but simply of the states being recorded, the underlying kinetic scheme is the same.

3. The term “rates” was often used instead of “rate constants”; this has been corrected.

4. We have fixed the spelling error pointed out by the Reviewer.

5. This point about the argument about our parameters fitting better than previous ones because previous parameters have been tuned to other data has been addressed in the

paragraph above.

6. The Reviewer pointed out that it is not clear how the model in equation (5) represents frequency/signals. We have added additional description to clarify that calcium injections are a part of the differential equation system and can be used to model calcium signals of different frequency or amplitude.

Moving on to the comments provided by Reviewer #2, the Reviewer wrote: “The fact that Equation 2 fits better than Equation 1 means that the strength of the laser pulse is completely random and cannot be controlled by specifying a PCD to a relevant degree. I think this should be clarified.”

We appreciate why the reviewer would draw this conclusion. With our approach, we do not claim that uncaging is completely independent of PCD; rather we use a single equation to capture both uncertainty in estimating the PCD, as well as other sources of variance. The amount of variance captured by the specified PCD value and finer points of calcium uncaging modeling are left for future research. We have included a statement to this effect in the revised version.

We addressed the feedback by Reviewer #2 concerning the usage of terminology and adjusted the relevant parts to follow Duchesne et al. (2021). We clarified and reduced the abuse of notation for the NLME parameters. We changed the symbol η to a more neutral symbol x to show that it is simply a description of a function. We clarified that the only place we used random effects is the calcium uncaging fraction. We clarified how the random effect prior distribution parameters were constrained. We expanded the section on model comparison, specifying that Laplace approximated marginal likelihood was used in AIC computations, described how and why we used Cohen’s d. We also addressed the further more minor notes provided by Reviewer #2, correcting the missing references, parentheses and other grammatical errors.

We again thank the Reviewers for their time, insightful feedback and contributing to the scientific community. We hope that our clarifications and revisions address their feedback satisfactorily.

Many thanks for your consideration,

Domas Linkevicius

---

## [Editor Report · Decision Letter 1]

21 Jan 2025

Fitting and comparison of calcium-calmodulin kinetic schemes to a common data set using non-linear mixed effects modelling

PONE-D-24-49899R1

Dear Dr. Linkevicius,

We’re pleased to inform you that your manuscript has been judged scientifically suitable for publication and will be formally accepted for publication once it meets all outstanding technical requirements.

Kind regards,

Pan Li, PhD

Academic Editor

PLOS ONE
---

## [Editor Report · Acceptance letter]

25 Jan 2025

PONE-D-24-49899R1 

PLOS ONE

Dear Dr. Linkevicius, 

I'm pleased to inform you that your manuscript has been deemed suitable for publication in PLOS ONE. Congratulations! Your manuscript is now being handed over to our production team.

Kind regards, 

on behalf of

Dr. Pan Li 

Academic Editor

PLOS ONE